# Dynamic Decision Learning: Test-Time Evolution for Abnormality Grounding in Rare Diseases

**Jun Li**[1 2] **Mingxuan Liu**[3] **Jiazhen Pan**[1 2] **Che Liu**[4] **Wenjia Bai**[4] **Cosmin I. Bercea**[* 1 2]
**Julia A. Schnabel**[* 1 2 5 6]

## Abstract

Clinical abnormality grounding for rare diseases is often hindered by data scarcity, rendering supervised fine-tuning infeasible and single-pass inference highly unstable. Thus, we propose *Dynamic Decision Learning (DDL)*, a framework that enables frozen LVLMs to refine their decisions across language and visual spaces by optimizing instructions and consolidating predictions under visual perturbations, thereby improving localization quality and producing a consensus-based reliability score that quantifies the model's confidence. Results on brain-imaging benchmarks, including a rare-disease dataset with 281 pathology types across 3B-72B models, show that *DDL* improves mAP@75 by up to 105% on rare-disease cases and surpasses adaptation baselines and supervised fine-tuning. Moreover, we show that *DDL* yields stronger calibration between consensus-based reliability scores and localization accuracy under severe distribution shifts and increasing task difficulty. The code is available at https://github.com/compai-lab/2026-ICML-DDL.

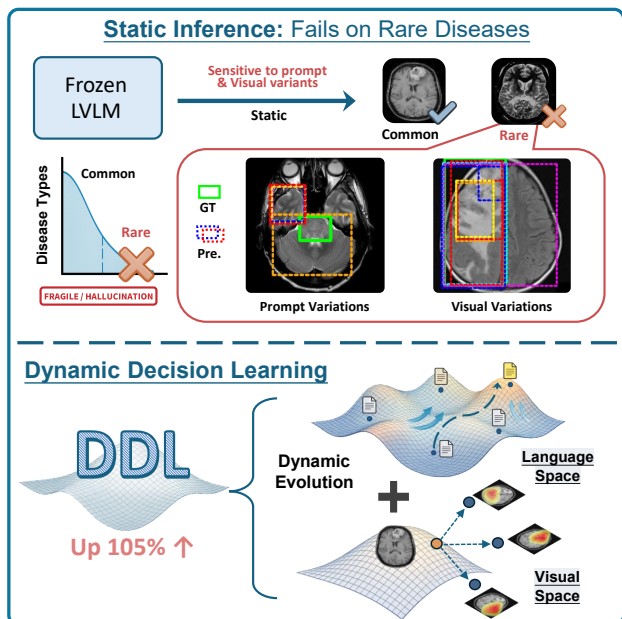

*Figure 1.* **Top**: Static inference with frozen LVLMs exhibits prompt and perturbation sensitivity on rare pathologies, leading to unstable and hallucinated localizations. **Bottom**: *DDL* performs test-time prompt optimization and multi-view verification, yielding substantially more stable and reliable localizations.

## 1. Introduction

Large vision-language models (LVLMs) have achieved strong performance on multimodal reasoning and visual grounding, including in medical imaging applications (Team et al., 2023; Achiam et al., 2023; Bannur et al., 2024; Ma et al., 2024; Zhu et al., 2024; Deperrois et al., 2025). Yet, under open-world clinical conditions characterized by long-tailed disease prevalence and domain shift, their abnormality localization remains unreliable, even for state-of-the-art models. In particular, grounding performance degrades sharply on rare and underrepresented pathologies (Bercea et al., 2025), limiting practical deployment in clinical decision support.

Most existing approaches address this limitation through domain-specific fine-tuning or post-training adaptation (Chen et al., 2024; Bannur et al., 2024; Guo et al., 2025; Deperrois et al., 2025). When sufficient annotated data is available, such methods can substantially improve grounding accuracy. However, they require repeated access to curated labels for each disease category, imaging protocol, and deployment environment. For rare and underrepresented pathologies, these assumptions do not hold in practice (Finlayson et al., 2021; Holste et al., 2024), making parameter-

---

[*]Shared senior authors. [1]Technical University of Munich [2]Munich Center for Machine Learning [3]University of Trento [4]Imperial College London [5]Helmholtz Munich [6]King's College London. Correspondence to: Jiazhen Pan <jiazhen.pan@tum.de, june.li@tum.de>.

*Proceedings of the 43rd International Conference on Machine Learning*, Seoul, South Korea. PMLR 306, 2026. Copyright 2026 by the author(s).

level adaptation difficult to scale under open-world clinical deployment. Rare clinical abnormalities are often indistinguishable from healthy tissue, requiring expensive expert verification that precludes large-scale data collection. Thus, acquiring sufficient labels for supervised training is practically impossible (Finlayson et al., 2021).

This lack of rare disease data also hinders the localization stability of LVLMs, leaving their predictions highly sensitive to minor variations in prompts and visual inputs (Teney et al., 2020; Farquhar et al., 2024). As shown in Figure 1, we find that grounding results shift substantially across semantically equivalent instructions or mild visual transformations. While genuine abnormalities remain spatially consistent across views, hallucinated detections are often visually unstable. These patterns suggest that a single-pass inference is fundamentally unreliable for long-tailed clinical tasks, as it fails to distinguish stable diagnostic signals from noise induced by these perturbations. In clinical settings, *a reliable diagnosis rarely stems from a single, static glance.* Instead, clinicians routinely cross-reference multiple views and iteratively re-evaluate findings to reduce diagnostic uncertainty (Berbaum et al., 1990; Croskerry, 2009; Waite et al., 2019). Current LVLM pipelines lack such verification, failing to distinguish stable clinical signals from noise.

To address this gap, we introduce *Dynamic Decision Learning (DDL)*, an inference procedure that formulates clinical grounding as hypothesis testing under controlled semantic and visual perturbations without parameter training. *DDL* decomposes this objective into two spaces. Within the language space, *DDL* reduces linguistic variability by optimizing instruction prompts to minimize empirical grounding risk over a development distribution. In the visual space, *DDL* defines perceptual reliability through the invariance of localized predictions under controlled perturbations and computes it by consolidating aligned hypotheses via structured matching. This yields detections paired with a consensus reliability score that reflects both recurrence and spatial consistency. Our main contributions are:

- We characterize semantic fragility and visual inconsistency as dominant inference-time failure modes in LVLM-based clinical grounding under long-tailed regimes.

- We introduce *Dynamic Decision Learning*, an inference-time framework that stabilizes grounding through prompt optimization and perturbation-based verification.

- We demonstrate consistent improvements over supervised and parameter-efficient adaptation across datasets, model scales, and rare disease distributions.

- We show that *DDL* systematically improves reliability calibration, with gains that increase as model capacity and task difficulty grow.

## 2. Related Works

**Clinical LVLMs and Parameter Adaptation.** Recent breakthroughs in LVLMs have catalyzed a shift toward generalist clinical assistants for medical reasoning (Arora et al., 2025; Sellergren et al., 2025), visual question answering (Chen et al., 2024; Kossen et al., 2024), and report generation (Li et al., 2022; Moor et al., 2023). The prevailing paradigm for adapting these models to medicine relies on data-driven parameter learning, primarily through supervised fine-tuning (SFT) or parameter-efficient updates like LoRA and its variants (Hu et al., 2022; Bannur et al., 2024; Wu et al., 2024; Deperrois et al., 2025; Pan et al., 2025). While effective in data-rich settings, these methods require repeated access to curated annotations and retraining for each deployment shift, which is impractical for rare diseases and evolving clinical distributions (Bercea et al., 2025; Zhao et al., 2025). In contrast, *DDL* avoids parameter updates and stabilizes grounding through inference-time verification.

**From Weight-based TTA to Test-Time Reasoning.** Traditional Test-Time Adaptation (TTA) focuses on mitigating domain shifts by updating model parameters or normalization statistics during inference (Wang et al., 2021; Sun et al., 2020; Liu et al., 2021; Zhang et al., 2022). Although successful in closed-set vision tasks, these methods are computationally expansive for large LVLMs and unstable under clinical noise (Sun et al., 2020). Recent work therefore explores inference-time reasoning and ouput-level optimization without modifying model weights (Shen et al., 2023; Nori et al., 2023; Long, 2023; Zheng et al., 2025). *DDL* follows this direction but focuses specifically on spatial grounding, using perturbation-based verification and structured matching to consolidate localization hypotheses.

**Instructional Evolution and Semantic Alignment.** The performance of LVLMs is highly sensitive to linguistic conditioning, motivating extensive work on prompt engineering and automated instruction optimization (Wei et al., 2022; Yang et al., 2023; Shanahan et al., 2023). Most existing methods target general-purpose LLMs via discrete search (Yang et al., 2023; Pryzant et al., 2023; Zhu et al., 2023) or learn continuous "soft" prompts for CLIP-based encoders (Zhou et al., 2022; Menon & Carl, 2023; Zhang et al., 2024). In contrast, clinical grounding necessitates high-level semantic alignment that captures both diagnostic logic and spatial priors (Li et al., 2026). *DDL* addresses this gap by optimizing prompts under empirical grounding risk and coupling them with perturbation-based visual verification.

**Hallucination and Self-Verification.** While LVLMs are increasingly powerful, they remain susceptible to hallucinations due to the inherent uncertainty in token-level generation (Li et al., 2023; Liu et al., 2024). Traditional calibration relies on predictive entropy or ensembles (Zhang et al., 2023;

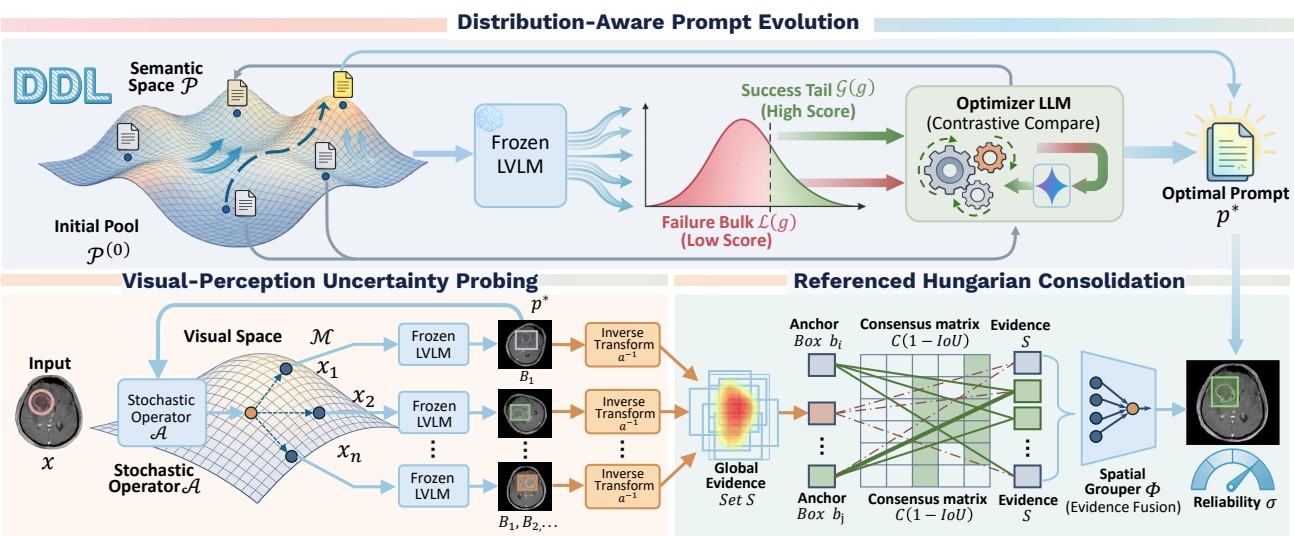

*Figure 2.* Overview of the Dynamic Decision Learning framework. *DDL* instantiates adaptive inference through two components: instruction-space optimization (DAPE), which refines task-specific prompts on a development set, and visual-consensus verification (V-PUP and RHC), which evaluates cross-view consistency and aggregates localization hypotheses via bipartite matching.

Farquhar et al., 2024), yet these metrics often fail to capture the spatial consistency. Recent verification methods utilize semantic consistency across multiple outputs as a reliability proxy (Kossen et al., 2024). However, such signals lack explicit grounding in geometric invariants. *DDL* addresses this by defining reliability through spatially aligned cross-view consensus. This approach couples uncertainty directly to grounding performance, ensuring localization hypotheses are verified against spatial consistency during inference.

## 3. Methodology

**Problem Formulation.** The goal of clinical abnormality grounding is to localize pathological regions in medical images that correspond to a natural language instruction. Formally, let $\mathbf{x} \in \mathcal{X}$ denote a brain scan and $\mathbf{p} \in \mathcal{P}$ a natural language instruction, e.g., "Identify all abnormalities in this scan." A frozen LVLM, denoted by $f_\theta$, processes the pair $(\mathbf{x}, \mathbf{p})$ to generate a set of candidate abnormality bounding boxes $\mathcal{B} = f_\theta(\mathbf{x}, \mathbf{p})$ in language token form.

### 3.1. *Dynamic Decisions Learning (DDL)*

We reformulate abnormality grounding from a static forward pass into an *adaptive inference strategy* $\pi \in \Pi$. A strategy specifies how the model explores and consolidates evidence across semantic and visual spaces. Our objective is to identify an optimal strategy $\pi^*$ that maximizes the expected decision utility $\mathcal{U}$:

$$\pi^* = \arg\max_{\pi \in \Pi} \mathbb{E}_{\mathbf{x} \sim \mathcal{D}} \left[ \mathcal{U} \left( \Phi \left( \{ f_\theta(\alpha(\mathbf{x}), \mathbf{p}) \}_{\alpha \sim \mathcal{A}, \, \mathbf{p} \sim \mathcal{P}_{opt}} \right) \right) \right],$$
(1)

where $\mathcal{U}$ combines grounding accuracy and reliability, $\Phi$ denotes a decision consolidation function, and $\mathcal{A}$ and $\mathcal{P}_{\text{opt}}$ denote distributions over visual perturbations and candidate instructions explored during prompt refinement, respectively. As detailed in the following sections, our *DDL* framework operationalizes this objective by integrating distribution-level instruction evolution with visual-consensus probing.

As illustrated in Fig. 2, DDL instantiates the adaptive inference objective in Eq. 1 through two complementary components. The first performs empirical optimization over instruction space to select prompts that minimize grounding error on a development distribution (Section 3.2). The second evaluates the stability of predicted regions under admissible visual transformations and aggregates consistent hypotheses through constrained matching (Section 3.3).

### 3.2. Instruction-Space Optimization (DAPE)

LVLM grounding performance is highly sensitive to instruction phrasing. To bridge this gap, we propose Distribution-Aware Prompt Evolution (DAPE), a module that discovers better instructional priors $\mathbf{p}^*$ by performing empirical optimization over the instruction space to identify prompts that minimize grounding error. DAPE maintains a performance-weighted history of candidate instructions, partitions them into high- and low-performing subsets, and uses an Optimizer LLM $\Psi$ as a semantic proxy to iteratively refine new candidates through contrastive feedback.

**Instruction Seeding and Partitioning.** The evolutionary loop begins by constructing an initial pool $\mathbb{P}^{(0)} = \{\mathbf{p}_1, \ldots, \mathbf{p}_N\}$ of size $N$ using the vanilla prompt and few-shot examples from $\mathcal{D}_{dev}$. This initialization ensures that

the population spans diverse styles prior to evaluation on the $\mathcal{D}_{dev}$. Throughout the search, we maintain a cumulative history $\mathcal{H}^{(g)} = \{(\mathbf{p}_i, y_i)\}_i$ of all candidates evaluated up to generation $g$ , where $y_i = \text{perf}(\mathbf{p}_i, \mathcal{D}_{dev})$ denotes the grounding performance (mAP) evaluated on the development set. This history is partitioned into two to model the success-failure landscape:

$$P(\mathbf{p} \mid y, \mathcal{H}^{(g)}) = \begin{cases} \mathcal{G}^{(g)}(\mathbf{p}) & \text{if } y \geq y^* \quad \text{(success)} \\ \mathcal{L}^{(g)}(\mathbf{p}) & \text{if } y < y^* \quad \text{(failure)} \end{cases} \quad (2)$$

where $y^*$ is a reference threshold (e.g., the performance of the vanilla prompt). By selecting the top-$k$ and bottom-$k$ performing instructions ($k \ll N$) to represent the success tail $\mathcal{G}^{(g)}$ and failure bulk $\mathcal{L}^{(g)}$, we provide $\Psi$ with explicit contrastive signals to identify robust linguistic patterns. Here, $k$ is a dynamic window (initially $k = 3$) that expands as the search history $\mathcal{H}$ grows, ensuring that the optimizer captures a representative distribution as the instruction pool scales.

**Contrastive Refinement.** At each iteration, $\Psi$ is tasked with synthesizing new candidates $\mathbf{p}^{(g+1)}$ by analyzing the semantic gap between the partitions:

$$\mathbf{p}^{(g+1)} \sim \Psi\left(\mathcal{G}^{(g)}, \mathcal{L}^{(g)}, \mathcal{K}\right). \quad (3)$$

where $\mathcal{K}$ specifies the task objective. Specifically, the optimizer LLM $\Psi$ adaptively switches its reasoning mode based on the distribution scores relative to the baseline $y^*$. If $\min(y \in \mathcal{L}^{(g)}) > y^*$, the optimizer performs *exploitative refinement* to amplify incremental success factors. Otherwise, $\Psi$ executes *contrastive compare* to isolate and suppress linguistic patterns that trigger specific failure modes. This distribution-aware feedback allows DAPE to perform a semantic descent toward the better prior $\mathbf{p}^*$ even under non-stationary search conditions. The process iterates until the performance in the success tail signals convergence.

### 3.3. Visual-Consensus Verification (V-PUP & RHC)

Beyond linguistic optimization, robust grounding requires that predicted regions remain stable under visual perturbations. The visual component of *DDL* estimates perceptual reliability by evaluating the invariance of localization hypotheses across transformed views.

**Visual-Perception Uncertainty Probing (V-PUP).** We exploit the observation that a genuine anatomical abnormality should remain spatially invariant under small visual perturbations, whereas spurious detections exhibit high variability. For a given image $\mathbf{x}$, we generate $M$ augmented views $\{\mathbf{x}_m = \alpha_m(\mathbf{x})\}_{m=1}^M$, where $\alpha \sim \mathcal{A}$ denotes a distribution of spatial-preserving transformations (e.g., slight rotations, or scaling).

The LVLMs processes each view using the optimized prompt $\mathbf{p}^*$ from the DAPE process:

$$\mathcal{B}_m = f_\theta(\mathbf{x}_m, \mathbf{p}^*), \quad m \in \{1, \dots, M\}. \quad (4)$$

This procedure yields a set of localization hypotheses that characterize the empirical variability of model predictions under perturbation.

**Referenced Hungarian Consolidation (RHC).** Aggregating these $M$ sets of multi-view predictions requires resolving spatial misalignment and suppressing correlated false positives. To this end, we propose *Referenced Hungarian Consolidation (RHC)*. We treat the prediction from the original image, $\mathcal{B}_{ref} = f_\theta(\mathbf{x}, \mathbf{p}^*)$, as a reference anchor. Since each view $m$ underwent a spatial transformation $\alpha_m$, we first apply the inverse transformation $\alpha_m^{-1}$ to map the predicted boxes back to the original coordinate space, yielding an *evidence set* $\mathcal{S} = \{\hat{\mathcal{B}}_m\}_{m=1}^M$, where $\hat{\mathcal{B}}_m = \alpha_m^{-1}(f_\theta(\mathbf{x}_m, \mathbf{p}^*))$.

To resolve spatial jitter and filter hallucinations, we construct a *consensus cost matrix* based on the negative Intersection-over-Union (-IoU) between the projected candidates and the reference anchor. We then solve a bipartite matching problem to align each $\hat{\mathcal{B}}_m$ with $\mathcal{B}_{ref}$ via the Hungarian algorithm. This enforces a strict one-to-one verification that prevents unrelated noise from inflating confidence.

Based on these cross-view alignments, we quantify the perceptual stability of each anchor box $b_j$ by defining the *Consensus Reliability Score* $\sigma_j \in [0, 1]$ as a weighted combination of *Consensus* (the recurrence of the signal) and *Consistency* (the spatial precision of the recurrence):

$$\sigma_j = \omega_1 \cdot \underbrace{\left(\frac{1 + \sum_m \mathbb{I}_{j,m}}{M + 1}\right)}_{\text{Consensus}} + \omega_2 \cdot \underbrace{\mathbb{E}_m[\text{IoU}(b_j, b_{j,m})]}_{\text{Consistency}} \quad (5)$$

where $\mathbb{I}_{j,m}$ is 1 if anchor $b_j$ finds a match in view $m$, and $b_{j,m}$ is the matched box, and $\mathbb{E}_m$ denotes the expectation over all successful matches. Here, $M + 1$ accounts for the total evidence pool (the original image plus $M$ augmented views). In practice, we set $\omega_1 = 0.6$ and $\omega_2 = 0.4$ to prioritize decision recurrence while penalizing spatial jitter.

Combining instruction-level optimization with visual-consensus verification, *DDL* produces a consolidated precition set $\mathcal{B}^* = \{(b_j, \sigma_j)\}$, where each detection is paired with an empirically calibrated reliability score. We formalize this unified inference protocol in Algorithm 1.

## 4. Experiments

**Brain Grounding Benchmarks.** We evaluate *DDL* on two clinical brain MRI benchmarks spanning common and long-tail pathology distributions: (i) BTD [Common Pathologies]

**Algorithm 1:** Dynamic Decisions Learning (DDL)

---

**Input:** Image $\mathbf{x}$, initial prompt $\mathbf{p}_0$, context $\mathcal{K}$, frozen LVLM
  $\quad f_\theta$, development set $\mathcal{D}_{dev}$
**Output:** Reliability-aware detections $\mathcal{B}^* = \{(b_j, \sigma_j)\}$
/* 1) Language Path: Prior Evolution (DAPE) */
Initialize history
  $\mathcal{H} \leftarrow \{(\mathbf{p}, y) \mid \mathbf{p} \in \mathbb{P}^{(0)}, \; y = \mathrm{perf}(\mathbf{p}, \mathcal{D}_{dev})\}$
**for** $g = 1$ **to** $G$ **do**
  $\quad (\mathcal{G}^{(g)}, \mathcal{L}^{(g)}) \leftarrow \mathrm{Partition}(\mathcal{H}, y^*)$  // Eq. (2)
  $\quad$ Sample prompt $\mathbf{p}^{(g)} \sim \Psi(\mathcal{G}^{(g)}, \mathcal{L}^{(g)}, \mathcal{K})$
  $\quad$ Evaluate $y^{(g)} \leftarrow \mathrm{perf}(\mathbf{p}^{(g)}, \mathcal{D}_{dev})$ and update $\mathcal{H}$

Select optimal prior $\mathbf{p}^* \leftarrow \arg\max_{(\mathbf{p}, y) \in \mathcal{H}} y$
/* 2) Visual Path: Consensus Verification (V-PUP & RHC) */
Obtain reference detections $\mathcal{B}_{ref} \leftarrow f_\theta(\mathbf{x}, \mathbf{p}^*)$
Generate augmented views $\{\mathbf{x}_m = \alpha_m(\mathbf{x})\}_{m=1}^M$, where
  $\alpha_m \sim \mathcal{A}$
**for** $m = 1$ **to** $M$ **do**
  $\quad \hat{\mathcal{B}}_m \leftarrow \alpha_m^{-1}(f_\theta(\mathbf{x}_m, \mathbf{p}^*))$  // Sec. 3.3
  $\quad \mathcal{M}_m \leftarrow \mathrm{Matching}(\hat{\mathcal{B}}_m, \mathcal{B}_{ref})$
/* 3) Reliability Scoring */
**for each** $b_j \in \mathcal{B}_{ref}$ **do**
  $\quad \sigma_j \leftarrow \mathrm{Score}(b_j, \{\mathcal{M}_m\}_{m=1}^M)$  // Eq. (5)
**return** $\mathcal{B}^* = \{(b_j, \sigma_j)\}$

---

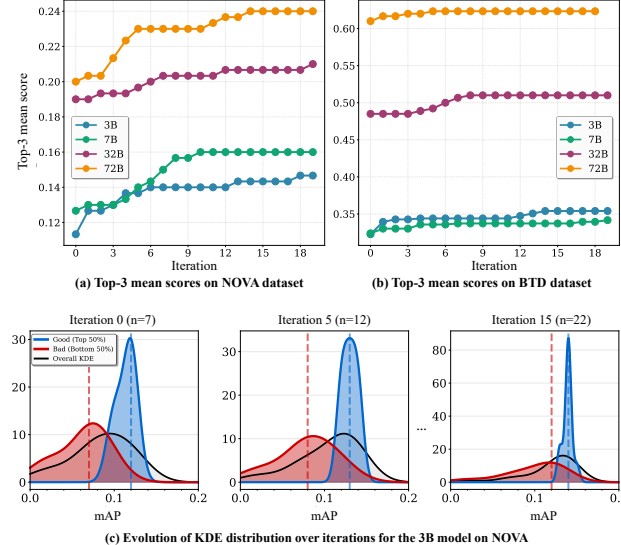

**(a) Top-3 mean scores on NOVA dataset**

**(b) Top-3 mean scores on BTD dataset**

**(c) Evolution of KDE distribution over iterations for the 3B model on NOVA**

*Figure 3.* Instructional optimization dynamics in DAPE. (a, b) Monotonic improvement in Top-3 candidate performance across model scales on the NOVA and BTD benchmarks. (c) Distributional shifts in the instruction pool: Kernel Density Estimation (KDE) reveals a rightward shift of the medium-score region and a sharpening of the high-performance tail, indicating convergence toward robust instructional priors.

(Darabi, 2022): a 3-class tumor dataset (glioma, meningioma, pituitary) and (ii) NOVA [Long-Tail OOD] (Bercea et al., 2025): a large-scale benchmark covering 281 rare pathologies and heterogeneous imaging protocols. NOVA serves as a rigorous out-of-distribution (OOD) stress test due to the extreme rarity and diversity of its findings. For each dataset, we reserve 100 samples for the development set ($\mathcal{D}_{dev}$) to optimize instructional priors and evaluate on the remaining held-out sets (293 for BTD; 806 for NOVA). We report mean average precision (mAP) at IoU thresholds of $\{0.25, 0.50, 0.75\}$ to assess localization precision across varying degrees of spatial strictness. Datasets statistics are provided in Appendix Sec. A.

**Implementation Details.** For the backbone LVLMs, we use the Qwen2.5-VL models ranging from 3B to 72B parameters (Bai et al., 2025) deployed via vLLM (Kwon et al., 2023). For instruction optimization (DAPE), we use a meta LLM (gemini-3-flash-preview) to generate candidate prompts on the development set. For visual verification (**V-PUP**), we sample $M = 7$ perturbed views using spatial-preserving transformations, including rotations, scaling, and flipping. For consolidation (**RHC**), we apply the Hungarian algorithm to match view-specific predictions to the reference anchor. Reliability scores are computed using $\lambda_1 = 0.6$ and $\lambda_2 = 0.4$. All results are averaged over three random seeds. Additional implementation details are provided in Appendix Sec. C and Sec. D.

**Compared Baselines.** We compare *DDL* against three classes of methods. (1) **Manual Prompting**, including

chain-of-thought (Wei et al., 2022), role-playing (Shanahan et al., 2023), visual description (Menon & Carl, 2023), strict constraints (Zhu et al., 2023), as well as recent prompting strategies combined with image strategies such as VISER (Izadi et al., 2025) and Visual-Instruct (Wang et al., 2025). (2) **Automated Optimizers**, including meta-optimization (Yang et al., 2023) and gradient-based optimization (Pryzant et al., 2023). (3) **Parameter Fine-tuning**: including full-model supervised fine-tuning (SFT) and LoRA (Hu et al., 2022) performed on the development set. All baselines are evluated under identical data and inference conditions. More details are provided in App. Sec. D.

We design our experiments to evaluate whether test-time adaptation improves abnormality grounding under long-tail clinical distributions. We analyze how instruction optimization and visual verification contribute to localization accuracy in Sec. 4.1, examine the emergence of calibration and reliability under increasing uncertainty and model scale in Sec. 4.2, and compare against prompt-bassed and training-based baselines in Sec. 4.3.

### 4.1. Analysis of *DDL* components

*1) DAPE iteratively shifts instruction distributions toward high-precision grounding.* We first analyze how DAPE modifies the distribution of candidate instructions during optimization. Figures 3(a,b) show monotonic improvements in Top-3 mean scores across model scales, indicating

*Table 1.* Decoding stochasticity (Language) vs. perceptual uncertainty (Visual) on NOVA. Bold indicates best performance per scale. Visual uncertainty scales more consistently with localization accuracy. Decisions are aggregated by simple averaging.

| Size | Method | mAP@25 | mAP@50 | mAP@75 |
|------|--------|--------|--------|--------|
| 3B | Language | 0.274 | 0.127 | 0.052 |
|    | Visual | **0.319** ▲ 16.6 | 0.122 ▼ 4.5 | 0.042 ▼ 19.5 |
| 7B | Language | 0.321 | 0.141 | 0.038 |
|    | Visual | **0.370** ▲ 15.2 | 0.160 ▲ 13.7 | 0.039 ▲ 2.9 |
| 32B | Language | 0.416 | 0.214 | 0.049 |
|     | Visual | **0.445** ▲ 7.1 | 0.218 ▲ 1.9 | 0.065 ▲ 32.6 |
| 72B | Language | 0.457 | 0.260 | 0.060 |
|     | Visual | **0.487** ▲ 6.7 | 0.273 ▲ 4.8 | 0.078 ▲ 30.2 |

*Table 2.* Consensus strategy comparison on NOVA. Baselines include Simple Average (SA), Weighted Average (WA), and DBSCAN clustering. Relative improvements ($\Delta$) are reported with respect to the SA baseline. RHC shows a strong trade-off towards high-precision alignment (mAP@75) across all model scales.

| Size | Method | mAP@25 | mAP@50 | mAP@75 |
|------|--------|--------|--------|--------|
| 3B | SA | 0.319 | 0.122 | 0.042 |
|    | WA | 0.292 | 0.148 | 0.057 |
|    | DBSCAN | **0.326** | **0.185** | 0.060 |
|    | RHC | 0.298 ▼ 6.5 | 0.150 ▲ 23.3 | **0.066** ▲ 58.1 |
| 7B | SA | **0.370** | 0.160 | 0.039 |
|    | WA | 0.360 | 0.187 | 0.063 |
|    | DBSCAN | 0.253 | 0.135 | 0.040 |
|    | RHC | 0.369 ▼ 0.2 | **0.206** ▲ 28.7 | **0.075** ▲ 93.0 |
| 32B | SA | 0.445 | 0.218 | 0.065 |
|     | WA | 0.445 | 0.208 | 0.065 |
|     | DBSCAN | 0.335 | 0.184 | 0.059 |
|     | RHC | **0.454** ▲ 2.0 | **0.266** ▲ 22.1 | **0.096** ▲ 47.6 |
| 72B | SA | 0.487 | 0.273 | 0.078 |
|     | WA | 0.493 | 0.300 | 0.101 |
|     | DBSCAN | 0.438 | 0.262 | 0.079 |
|     | RHC | **0.500** ▲ 2.6 | **0.301** ▲ 10.5 | **0.107** ▲ 36.1 |

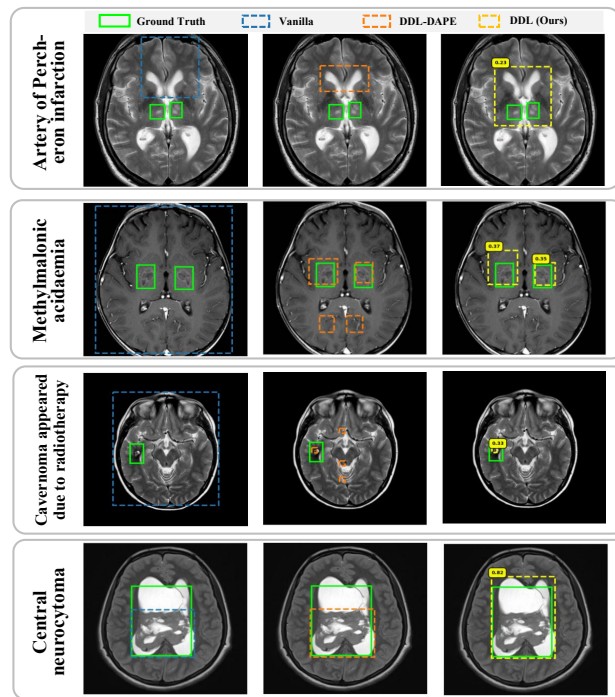

*Figure 4.* **Qualitative grounding results on rare pathologies from NOVA.** Blue dashed boxes denote the vanilla baseline, orange dashed boxes correspond to *DDL-DAPE*, and yellow dashed boxes show the final *DDL* output. Green solid boxes indicate ground truth. *DDL* progressively suppresses unstable detections and improves spatial alignment with ground truth.

that DAPE consistently identifies more effective instruction prompts. The kernel density estimates in Fig. 3-c further show that optimization does not simply uncover isolated high-scoring outliers, but progressively shifts and sharpens the entire candidate distribution. Both the median and upper tail of the distribution increase over iterations, reflecting a progressive concentration of probability mass in regions of instruction space associated with reliable localization. These trends indicate that DAPE performs structured exploration and exploitation in the discrete prompt space, yielding better and stable instruction prompts. More results are provided in Appendix Fig. 9 for improvements on the development set, and Fig. 10–13 for prompt evolution examples.

*2) Visual uncertainty offers a more reliable grounding signal than linguistic uncertainty.* We compare uncertainty estimates derived from high-temperature decoding ($T = 1.0$) with those obtained through multi-view perturbation. As shown in Table 1, visual uncertainty consistently surpasses

language-level uncertainty across all model scales. This indicates that grounding accuracy is more strongly associated with perceptual stability than with variability in token-level likelihoods from language.

High-temperature sampling introduces substantial stochasticity in token generation, leading to unstable spatial hypotheses. In contrast, V-PUP probes the consistency of localized predictions under controlled geometric transformations. For the 3B model, this perturbation-based evaluation reveals significant performance degradation at strict thresholds (mAP@75), exposing reliance on fragile visual cues. Larger models (32B+) exhibit substantially greater invariance, maintaining or even improving localization precision under perturbation. These findings confirm that V-PUP effectively separates robust spatial evidence from stochastic decoding noise. Appendix Table 11 provides additional results using the vanilla prompt and weighted-average fusion.

*3) Structured matching improves spatial alignment at strict localization thresholds.* RHC formulates cross-view consensus as a bipartite assignment problem anchored to a reference prediction. This contrasts with reference-free aggregation methods such as simple averaging (SA) and DBSCAN, which rely on heuristic clustering. As shown in table 2, reference-free methods achieve slightly higher

*Table 3.* Scaling behavior of DDL across model sizes from 3B to 72B. Performance is evaluated on both the NOVA and BTD datasets, with relative improvements ($\Delta$) reported over the Vanilla baseline. Results show that DDL consistently improves grounding accuracy across all model scales, with particularly large gains at stricter localization thresholds, reaching up to 104% on mAP@75.

| D. | Size | Method | mAP@25 | mAP@50 | mAP@75 |
|---|---|---|---|---|---|
| NOVA | 3B | Vanilla | 0.221 | 0.088 | 0.040 |
| | | DDL (Ours) | 0.298 ▲35.0 | 0.150 ▲69.6 | 0.066 ▲66.0 |
| | 7B | Vanilla | 0.286 | 0.135 | 0.036 |
| | | DDL (Ours) | 0.369 ▲29.1 | 0.206 ▲52.4 | 0.075 ▲104.7 |
| | 32B | Vanilla | 0.406 | 0.208 | 0.058 |
| | | DDL (Ours) | 0.454 ▲11.8 | 0.266 ▲28.1 | 0.096 ▲65.5 |
| | 72B | Vanilla | 0.411 | 0.245 | 0.065 |
| | | DDL (Ours) | 0.500 ▲21.7 | 0.301 ▲22.8 | 0.107 ▲65.4 |
| BTD | 3B | Vanilla | 0.418 | 0.370 | 0.269 |
| | | DDL (Ours) | 0.403 ▼3.5 | 0.379 ▲2.5 | 0.283 ▲5.5 |
| | 7B | Vanilla | 0.348 | 0.289 | 0.154 |
| | | DDL (Ours) | 0.383 ▲10.1 | 0.302 ▲4.4 | 0.159 ▲3.7 |
| | 32B | Vanilla | 0.496 | 0.367 | 0.160 |
| | | DDL (Ours) | 0.602 ▲21.4 | 0.433 ▲17.9 | 0.206 ▲28.7 |
| | 72B | Vanilla | 0.571 | 0.488 | 0.306 |
| | | DDL (Ours) | 0.650 ▲13.9 | 0.572 ▲17.1 | 0.346 ▲13.1 |

mAP@25 on smaller models, e.g., 0.319 vs. 0.298 for 3B, due to permissive aggregation of loosely aligned hypotheses. However, this permissiveness degrades performance under strict localization criteria. By enforcing one-to-one matching with a canonical anchor, RHC suppresses spatial drift and correlated noise across views. This leads to substantially stronger performance at high-precision thresholds, including a 93.0% relative improvement in mAP@75 for the 7B model. These results indicate that explicit structural constraints are essential for preserving spatial consistency under multi-view aggregation.

**4) Test-time refinement of localization hypotheses stabilizes grounding.** Fig. 4, shows how *DDL* progressively improves localization through iterative verification. Single-pass inference (blue) often produces incomplete or diffuse detections, missing subtle abnormalities, or generates overly broad regions. Instruction refinement via DAPE (orange) improves semantic focus but remains susceptible to visually induced false positives in complex scans. *DDL* enforces consistency by evaluating predictions across perturbations and consolidating them through structured matching. As a result, *DDL* produces localized predictions that closely match true anatomical abnormalities.

**5) DDL narrows the performance gap across model scales.** Table 3 shows consistent improvements from *DDL* on the challenging NOVA benchmark across all model sizes. Notably, the 32B model exceeds the zero-shot 72B baseline at all evaluated thresholds (e.g., 0.454 vs. 0.411 mAP@25).

On the BTD dataset, smaller models (3B) exhibit a modest reduction in coarse recall (mAP@25) due to strict consolidation, but achieve substantial gains at high-precision thresholds (mAP@75).

These results indicate that test-time refinement partially compensates for limited model capacity by improving localization precision. Additional examples across model scales are provided in Appendix Fig. 10–13.

*Table 4.* Reliability score calibration across datasets and model scales. Metrics include average IoU ($\overline{\text{IoU}}$), mean reliability score ($\overline{\sigma}$), score standard deviation (std($\sigma$)), mean absolute error (**MAE** $= \mathbb{E}[|\sigma - \text{IoU}|]$), and Pearson correlation ($r$) between the reliability score $\sigma$ (defined in Eq. 5) and IoU. Significance ($Sig$) levels: $^{**}p < 0.01, ^{***}p < 0.001$, ns: not significant.

| D. | S. | $\overline{\text{IoU}}$ | $\overline{\sigma}$ | std($\sigma$) | MAE | $r$ | $Sig$ |
|---|---|---|---|---|---|---|---|
| NOVA Hard | 3B | 0.231 | 0.747 | 0.179 | 0.532 | 0.096 | ** |
| | 7B | 0.254 | 0.624 | 0.209 | 0.395 | 0.331 | *** |
| | 32B | 0.314 | 0.632 | 0.199 | 0.341 | 0.528 | *** |
| | 72B | 0.357 | 0.680 | 0.191 | 0.340 | 0.535 | *** |
| BTD Easy | 3B | 0.357 | 0.803 | 0.147 | 0.492 | 0.052 | ns |
| | 7B | 0.309 | 0.715 | 0.180 | 0.467 | -0.037 | ns |
| | 32B | 0.419 | 0.705 | 0.193 | 0.310 | 0.622 | *** |
| | 72B | 0.501 | 0.733 | 0.200 | 0.282 | 0.597 | *** |

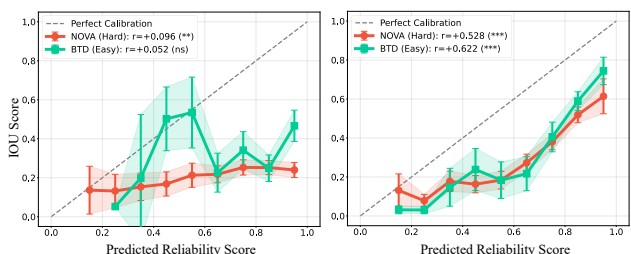

*Figure 5.* Spatial calibration dynamics on the BTD and NOVA datasets. (Left) The 3B model exhibits flat, decoupled confidence-performance curves ($r \approx 0.1$). (Right) Model scaling (32B) unlocks a strong alignment with the diagonal, transforming consensus reliability into a predictive indicator of clinical grounding success.

### 4.2. Scaling Law

**1) DDL calibration emerges with model size.** Table 4 shows that, under the proposed *DDL* inference scheme, the correlation between the reliability score $\sigma$ and localization accuracy (IoU) increases monotonically with model scale on NOVA, from $r = 0.096$ (3B) to $r = 0.535$ (72B). This improvement is accompanied by a systematic reduction in calibration error, with mean absolute error decreasing from 0.532 to 0.340. Above 32B parameters, the alignment becomes statistically significant ($p < 0.001$), with $r > 0.5$ across both benchmarks.

These results indicate that reliable internal uncertainty estimation is not present in small models but emerges as a function of scale. Larger models produce internally consistent

*Table 5.* Comparative analysis against training-free baselines across common (BTD) and rare (NOVA) clinical scenarios on the 3B model. We evaluate mAP at various IoU thresholds. Baselines include: CoT (Wei et al., 2022), Visual-Des. (Menon & Carl, 2023), Role-Play (Shanahan et al., 2023), Strict Const. (Zhu et al., 2023), VISER (Izadi et al., 2025), Visual-Instruc. (Wang et al., 2025), Meta Opt. (Yang et al., 2023), and Gradient Opt. (Pryzant et al., 2023). Bold indicates best zero-shot performance; relative percentage changes versus Vanilla are shown in subscript.

| D. | Method | mAP@25 | mAP@50 | mAP@75 |
|---|---|---|---|---|
| NOVA | Vanilla | 0.221 | 0.088 | 0.040 |
| | CoT | 0.092 ▼58.2 | 0.019 ▼78.6 | 0.003 ▼91.5 |
| | Visual-Des. | 0.197 ▼10.9 | 0.067 ▼24.2 | 0.028 ▼30.0 |
| | Role-Play | 0.226 ▲2.1 | 0.094 ▲6.2 | 0.038 ▼4.5 |
| | Strict Const. | 0.244 ▲10.3 | 0.113 ▲27.8 | 0.049 ▲23.0 |
| | VISER | 0.239 ▲8.3 | 0.100 ▲12.7 | 0.038 ▼4.5 |
| | Visual-Instruc. | 0.036 ▼83.9 | 0.010 ▼88.9 | 0.002 ▼95.8 |
| | Gradient Opt. | 0.039 ▼82.5 | 0.009 ▼89.7 | 0.003 ▼91.5 |
| | Meta Opt. | 0.282 ▲27.5 | 0.121 ▲37.5 | 0.049 ▲22.3 |
| | **DDL (Ours)** | **0.298** ▲35.0 | **0.150** ▲69.7 | **0.066** ▲66.0 |
| BTD | Vanilla | 0.418 | 0.370 | 0.269 |
| | CoT | 0.256 ▼38.7 | 0.171 ▼53.9 | 0.055 ▼79.7 |
| | Visual-Des. | 0.340 ▼18.5 | 0.289 ▼21.8 | 0.220 ▼18.2 |
| | Role-Play | 0.355 ▼15.0 | 0.304 ▼17.8 | 0.230 ▼14.4 |
| | Strict Const. | 0.337 ▼19.4 | 0.297 ▼19.7 | 0.209 ▼22.1 |
| | VISER | 0.412 ▼1.3 | 0.365 ▼1.2 | 0.257 ▼4.3 |
| | Visual-Instruc. | 0.191 ▼54.3 | 0.123 ▼66.7 | 0.083 ▼69.1 |
| | Gradient Opt. | 0.188 ▼55.0 | 0.180 ▼51.4 | 0.124 ▼53.8 |
| | Meta Opt. | 0.408 ▼2.2 | 0.357 ▼3.4 | 0.269 0.0 |
| | **DDL (Ours)** | **0.403** ▼3.5 | **0.379** ▲2.5 | **0.283** ▲5.5 |

reliability signals, whereas smaller models yield scores that remain weakly coupled to true localization performance.

*2) Calibration collapse on easy tasks in small models.* Under the proposed *DDL* inference scheme, small models exhibit poor calibration on simpler task distributions. On the BTD benchmark, the 3B model shows weak and unstable alignment between reliability scores and localization accuracy (Fig. 5, left). Table 4 confirms this behavior, with statistically insignificant correlations for both 3B ($r = 0.052$) and 7B ($r = -0.037$) models. In contrast, on the more challenging NOVA benchmark, the same models exhibit measurable alignment (e.g., $r = 0.096$, $p < 0.01$ for 3B), indicating that reliability estimates become informative only under increased task difficulty. Larger models (32B+) maintain stable calibration across both benchmarks (Fig. 5, right). Full results are provided in Appendix Fig. 8–7.

### 4.3. Comparison with other baselines

*1) DDL consistently outperforms prompt optimization methods.* Table 5 compares *DDL* with a broad range of manual and automated prompting strategies. While automated optimizers such as Meta-Opt improve over manual prompting (e.g., increasing mAP@25 from 0.221 to 0.282 on NOVA), their performance remains substantially below *DDL*, particularly at strict localization thresholds.

*Table 6.* Comparison of DDL with training-based adaptation methods (LoRA (Hu et al., 2022), SFT (Zheng et al., 2024)) on the 3B model using the development set for training. DDL generalizes better to long-tail distributions, demonstrating that supervised fine-tuning remains sub-optimal for rare disease grounding.

| D. | Method | mAP@25 | mAP@50 | mAP@75 |
|---|---|---|---|---|
| NOVA | Vanilla | 0.221 | 0.088 | 0.040 |
| | LoRA | 0.225 ▲1.6 | 0.092 ▲3.7 | 0.042 ▲3.8 |
| | SFT | 0.283 ▲28.1 | 0.128 ▲45.0 | 0.046 ▲16.0 |
| | **DDL (Ours)** | **0.298** ▲35.0 | **0.150** ▲69.7 | **0.066** ▲66.0 |
| BTD | Vanilla | 0.418 | 0.370 | 0.269 |
| | LoRA | 0.429 ▲2.7 | 0.379 ▲2.5 | 0.277 ▲3.0 |
| | SFT | **0.479** ▲14.7 | **0.413** ▲11.7 | **0.308** ▲14.8 |
| | **DDL (Ours)** | 0.403 ▼3.5 | 0.379 ▲2.5 | 0.283 ▲5.5 |

On the challenging NOVA benchmark, *DDL* outperforms all prompting-based baselines, including reasoning-based and optimization-based methods, by large margins at mAP@75. These results show that improved instruction design alone cannot resolve the localization failures induced by perceptual instability. Reliable grounding on rare pathologies requires explicit test-time verification and consolidation.

*2) DDL outperforms supervised fine-tuning on rare diseases.* On NOVA, supervised fine-tuning fails to translate training gains into reliable localization (Table 6). In contrast, *DDL* delivers consistent improvements across all thresholds, outperforming SFT at both coarse and strict criteria (0.298 vs. 0.283 mAP@25; 0.066 vs. 0.046 mAP@75). These results indicate that, under extreme data sparsity, stabilizing inference is more effective than optimizing parameters. See Appendix Table 10 for additional experiments comparing the baselines across different model sizes.

## 5. Conclusion and Discussion

We introduced *Dynamic Decisions Learning (DDL)*, an inference-time framework for abnormality grounding with frozen LVLMs. *DDL* combines distribution-level instruction optimization with test-time multi-view verification to stabilize localization under long-tail and out-of-distribution conditions. Across two brain MRI benchmarks and models from 3B to 72B parameters, *DDL* substantially improves high-precision localization and consistently outperforms supervised fine-tuning in rare-disease settings.

Beyond accuracy, our experiments show that *DDL* induces well-calibrated reliability estimates that increasingly align with localization performance as model scale grows. This reveals a systematic relationship between model capacity, test-time verification, and spatial calibration. In particular, large models with *DDL* exhibit reliable internal consistency signals, while smaller models remain weakly calibrated. These findings position test-time optimization as a viable alternative to weight adaptation for open-world medical imaging

where data scarcity and distribution shifts are prevalent.

More broadly, *DDL* illustrates how adaptive inference can complement large pretrained models in long-tail settings. A remaining limitation of *DDL* is its increased test-time computation. Future work will focus on amortizing verification, extending the framework to additional modalities, and integrating learned uncertainty models. This work motivates the systematic study of inference-time adaptation as a core component of reliable multimodal systems.

## Acknowledgments

We thank Prof. Dr. Benedikt Wiestler for his contributions to the NOVA dataset. This work was supported by the Munich Center for Machine Learning (MCML). C.I.B. was funded via the EVUK program ("Next-generation AI for Integrated Diagnostics") of the Free State of Bavaria.

## Impact Statement

This work introduces a dynamic decision learning framework for abnormality grounding in rare-disease imaging, aiming to improve the reliability of current LVLMs. The proposed approach raises no notable ethical concerns regarding its motivation, design, implementation, or data usage. All datasets employed in this study are publicly available and de-identified. As a methodological contribution intended for decision-support rather than autonomous diagnosis, DDL does not pose foreseeable societal risks beyond those commonly associated with clinical AI research.

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

# Appendix

**Contents**

# A. Dataset Details

## A.1. NOVA: Rare diseases distribution

The NOVA benchmark presents a significant challenge for clinical grounding due to its exceptional pathological diversity. It encompasses 281 unique rare diseases, leading to highly heterogeneous abnormal regions across various MRI modalities and scan orientations. As illustrated in Fig. 6, we visualize the taxonomic distribution of pathologies in the NOVA dataset, categorized by neuroradiological types. This distribution underscores the extreme diversity of the benchmark.

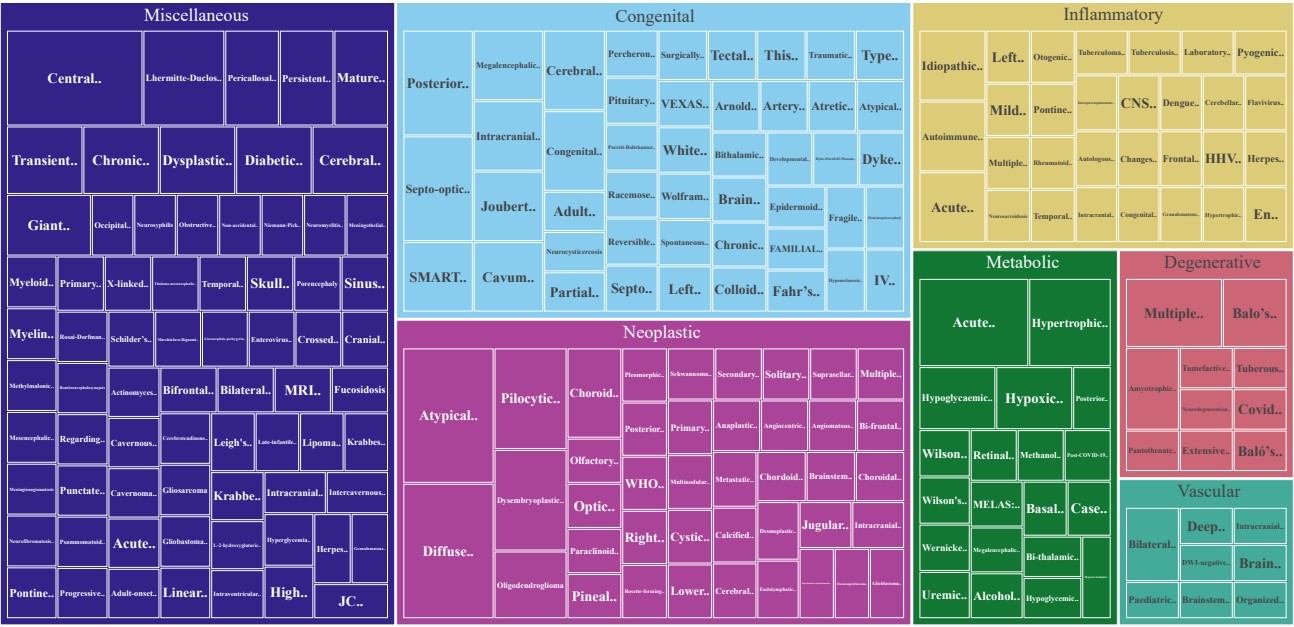

*Figure 6.* Treemap visualizing the taxonomic distribution of clinical pathologies in the NOVA dataset.

## A.2. BTD: Brain tumor dataset

The Brain Tumor Dataset (BTD) (Darabi, 2022) is utilized as a comparative baseline representing common clinical pathologies. It comprises multi-modal brain MRI scans categorized into three primary tumor types: *pituitary tumors*, *meningioma*, and *glioma*. Unlike the rare diseases in NOVA, these categories represent high-frequency pathological patterns with well-established radiological features. Our evaluation is conducted on a held-out test set of 293 samples, the pathological distribution of which is detailed in Table 7. These common pathologies provide a fundamental reference point for assessing how clinical grounding models handle standard diagnostic tasks. By evaluating on BTD, we can distinguish between a model's general localization capability and its robustness to the extreme OOD distribution shifts encountered in the NOVA benchmark.

*Table 7.* Distribution of the BTD (293 samples).

| Tumor Type | Count |
| --- | --- |
| Pituitary Tumor | 124 |
| Meningioma | 91 |
| Glioma | 78 |

# B. Notations

In this section, we formalize the variables and operators used throughout the DDL methodology. As detailed in the problem formulation (Sec. 3), we define abnormality grounding as an adaptive inference trajectory $\pi$ that maximizes the expected decision utility $\mathcal{U}$ across both semantic and visual spaces. Specifically, DDL operationalizes the objective of identifying an optimal strategy $\pi^*$ that integrates evidence from multi-view visual perspectives $\mathcal{A}$ and evolved instructional priors $\mathbf{p}^*$. The framework leverages the synergy between the Language Path (DAPE) for distribution-level semantic alignment and the Visual Path (V-PUP and RHC) for instance-level perceptual stability and consensus verification. Table 8 summarizes the key notations and definitions used throughout the DDL framework.

*Table 8.* Summary of Notations and Definitions

| Notation | Definition |
|---|---|
| $\mathbf{x} \in \mathcal{X}, \mathbf{p} \in \mathcal{P}$ | Input brain MRI scan and natural language instruction. |
| $f_\theta$ | Frozen Large Vision-Language Model (LVLM) parameters. |
| $\mathcal{B} = \{b_1, \ldots, b_n\}$ | Set of predicted abnormality bounding boxes ($b_j = [x_1, y_1, x_2, y_2]$). |
| $\pi \in \Pi$ | Adaptive inference strategy (Dynamic Decisions trajectory). |
| $\mathcal{U}$ | Decision utility function combining localization precision and reliability. |
| $\Phi$ | Decision consolidation function (Referenced Hungarian Consolidation). |
| $\Psi$ | Meta-Optimizer LLM used for instruction evolution (DAPE). |
| $\mathbb{P}^{(g)}$ | Instruction pool at evolutionary iteration $g$. |
| $\mathcal{H}^{(g)}$ | Cumulative performance history of instructions on the dev set $\mathcal{D}_{dev}$. |
| $\mathcal{G}^{(g)}, \mathcal{L}^{(g)}$ | Success tail and failure bulk of the instruction distribution. |
| $y^*$ | Performance baseline (vanilla prompt score). |
| $\mathcal{A}$ | Distribution of spatial-preserving visual transformations. |
| $M$ | Number of stochastic view-probing samples (augmentations). |
| $\alpha_m, \alpha_m^{-1}$ | $m$-th stochastic spatial transformation and its corresponding inverse. |
| $\mathcal{B}_{ref}$ | Reference anchor detections generated from the original image $\mathbf{x}$. |
| $\hat{\mathcal{B}}_m$ | Candidates from view $m$ projected back into the reference coordinates. |
| $\sigma_j$ | Consensus Reliability Score for detection $b_j$. |
| $\omega_1, \omega_2$ | Relative weights for consensus recurrence and spatial consistency. |

# C. Technical Details

This section provides comprehensive implementation details for the three core components of the DDL framework: Distribution-Aware Prompt Evolution (DAPE), Visual-Perception Uncertainty Probing (V-PUP), and Referenced Hungarian Consolidation (RHC).

## C.1. Details of the Distribution-Aware Prompt Evolution

The Language Path utilizes **DAPE** to discover optimal instructional priors through an iterative evolutionary search.

**Instruction Seeding and Partitioning.** The initial population $\mathbb{P}^{(0)}$ is seeded with a baseline clinical instruction (see *Vanilla Prompt* below). For partitioning, we maintain a history $\mathcal{H}$ of all evaluated prompts. At each generation $g$, we perform **Success-Failure Partitioning** by selecting the Top-$k$ prompts as the success tail $\mathcal{G}^{(g)}$ and the Bottom-$k$ prompts as the failure bulk $\mathcal{L}^{(g)}$. Initially $k = 3$, and $k$ gradually increases by 1 through iterations. This contrastive pair allows the optimizer to identify distribution differences and learn from them to generate improved prompts that correlate with higher mAP.

---

**Vanilla Prompt ($\text{p}_{\text{vanilla}}$)**

```
Return bounding boxes of any abnormal areas as JSON format:
If the image do not have the target, return the string:  "no target".
If detected, return a list of 2D bounding boxes around the target regions:

[
{"bbox_2d":  [x1, y1, x2, y2], "label":  "label"},
...
]
```

---

**Meta-Optimizer Initialization.** To generate the initial population $\mathbb{P}^{(0)}$ with linguistic diversity, we employ a initialization prompt $\Psi_{\text{init}}$ that asks the Meta-LLM to create $N = 5$ variants of the vanilla instruction. These variants incorporate different prompt engineering strategies (e.g., chain-of-thought reasoning, explicit output formatting) while staying semantically close to the vanilla prompt. The initialization prompt is shown in the box below:

---

**Meta-Optimizer Initialization Prompt ($\Psi_{\text{init}}$)**

```
You are an expert prompt engineer for medical imaging tasks.

Task Context:  Brain MRI Abnormality Grounding
Your goal is to improve instructions for detecting and localizing pathological
abnormalities in brain MRI scans.  The model must identify abnormalities
in the image with precise bounding box coordinates.

Given the vanilla instruction below, generate 5 diverse but semantically
equivalent variations that incorporate different prompt strategies.

Each variation should:
(1) Maintain the core abnormality grounding task
(2) Use different linguistic patterns or reasoning approaches
(3) Ensure JSON output format compliance

Vanilla Instruction:
``[VANILLA_INSTRUCTION]''

Output Format (JSON only):

{
"variant_1":  "...",
"variant_2":  "...",
"variant_3":  "...",
"variant_4":  "...",
"variant_5":  "..."
}
```

**Generated Prompt Examples.** During initialization, the Meta-LLM ($\Psi$) produces diverse variants of the vanilla prompt. Representative examples are shown side-by-side below:

**Variant 1:**

```
Analyze this MRI image step-by-step.
1.  Check for symmetry and identify
asymmetries.
2.  Look for abnormal signal
intensities.
3.  Identify any mass effects or
distortions.

Return bounding boxes in JSON format:
[
{"bbox_2d":  [x1, y1, x2, y2],
"label":  "abnormality"}
]
```

**Variant 2:**

```
Act as an expert neuroradiologist.
Carefully examine this MRI for any
clinically
significant abnormalities.  Flag
regions with high
confidence of being a lesion, tumor,
or infarct.

Return results strictly as JSON:
[
{"bbox_2d":  [x1, y1, x2, y2],
"label":  "pathology_type"}
]
```

**Success-Failure Partitioning and Contrastive Refinement.** DAPE operationalizes the acquisition of $\mathbf{p}^*$ by treating the Meta-Optimizer $\Psi$ as a density estimator over the prompt space $\mathcal{P}$. In each iteration $g$, we employ a **Dynamic Window Partitioning** strategy. We define the success tail $\mathcal{G}^{(g)}$ as the $k$ highest-performing prompts and the failure bulk $\mathcal{L}^{(g)}$ as the $k$ lowest-performing prompts from the history $\mathcal{H}^{(g)}$. To balance exploration and exploitation, the window size $k$ is defined based on the iteration progress: $k = \min(\lfloor g/2 \rfloor + 1, |\mathcal{H}|/2)$. This ensures that early iterations focus on high-variance trajectories, while later iterations leverage a broader distribution of historical evidence.

**Logic of the Meta-Optimizer ($\Psi$).** The refinement instructions for $\Psi$ are dynamically adjusted based on the population's performance relative to the vanilla baseline $y_{\text{vanilla}}$ to ensure productive evolution:

---

**Meta-Optimizer Refinement (Contrastive Scenario)**

```
You are analyzing prompts for medical image analysis tasks.  Your task is to
understand what makes some prompts work better and others worse...
HIGH PERFORMANCE PROMPTS (What Works Well):  [Success Tail]
LOW PERFORMANCE PROMPTS (What Doesn't Work):  [Failure Bulk]

YOUR TASK:
1.  Analyze key differences between high and low performing prompts.
2.  Identify specific successful elements to incorporate.
3.  Understand what aspects in low-performance prompts hurt results.
4.  Generate ONE improved prompt avoiding identified pitfalls.
5.  The output prompt should also like the success prompt, avoid too long.
Output Format:  <IMPROVED_PROMPT> [Your Prompt] </IMPROVED_PROMPT>
```

---

**Meta-Optimizer Refinement (Exploitative Scenario)**

```
You are analyzing prompts for medical image analysis tasks.  All candidate prompts
perform better than the baseline, so analyze what makes them successful and generate
an even better version.
BASE PROMPT (Baseline Reference):  [Base Score & Text]
BEST PERFORMING PROMPTS: [Top-k Variants & Scores]
RELATIVELY WEAKER PROMPTS: [Bottom-k Variants & Scores]

YOUR TASK:
1.  Analyze what makes the best prompts work so well.
2.  Identify the key success factors in both prompt sets.
3.  Understand what distinguishes the best from the weaker ones.
4.  Generate ONE improved prompt combining the best elements.
Output strictly using <ANALYSIS> and <IMPROVED_PROMPT> tags.
```

**Convergence Criteria via Top-$k$ Stability.** We monitor the convergence of the DAPE process by calculating the standard deviation (std) of the Top-3 prompts. We define the search as converged when std(Top-3) $< 10^{-4}$, signaling that the Meta-LLM is no longer receiving a meaningful "textual gradient" to further distinguish candidate instructions.

**Meta-Optimizer Configuration ($\Psi$).** We employ the Gemini 3 Flash (Preview) model, accessed via the *OpenRouter* API, as the Meta-LLM for both initialization and iterative refinement. The temperature is set to 0.7 with $top\_p = 0.9$ to ensure sufficient exploration of the instruction space. During the evolutionary search, the optimizer receives contrastive sets $\{\mathcal{G}^{(g)}, \mathcal{L}^{(g)}\}$ to perform the language prompt optimization.

**Target Model Deployment and Evaluation.** To support the iterative nature of DAPE and the online verification of V-PUP, target LVLMs are deployed using the `vLLM` engine. We utilize a consistent batch size of 32 across all inference tasks to balance throughput and memory stability. Models are executed with `bfloat16` precision across NVIDIA H800 GPUs.

### C.2. Visual-Consensus Verification

The visual path ensures reliability through multi-view sanity checks. Our implementation handles the transition between transformed visual spaces and the reference coordinate system, ensuring alignment with the objective in Eq. 1.

**Visual-Perception Uncertainty Probing (V-PUP).** Following the formulation in Sec. 3.3, we generate 7 stochastic perturbations of the input image: random rotations ($\pm 3°$), random scaling ($\times 0.9, \times 1.1$), random translations ($\pm 20$ pixels), and horizontal flipping. Together with the original unperturbed image, these constitute a total evidence pool of $M_{total} = 8$ views. To optimize throughput, we utilize the `vLLM` engine to process all 8 views in parallel within a single inference batch.

**Coordinate Normalization and Inverse Mapping.** The `Qwen2.5-VL-Instruct` series typically outputs detection coordinates directly in the pixel space of the input image. As implemented in our pipeline, we ensure that these coordinates correspond to the dimensions $(H_m, W_m)$ of each specific augmented view. After inference, we apply the inverse affine

transform $\mathbf{M}_m^{-1}$ to map all candidate boxes from the $M$ views back to the reference coordinate system $(x, y)_{ref}$ of the original scan. This step is critical for the RHC matching process, as visual perturbations alter the absolute position and scale of pathological features relative to the image borders.

### C.3. Referenced Hungarian Consolidation

To resolve spatial jitter and filter hallucinations (Eq. 5), we treat the original view detection $\mathcal{B}_{ref}$ as the definitive anchor. We then construct a bipartite graph where one side contains anchor boxes from the original view and the other side contains detection hypotheses from augmented views (inverse-transformed to original coordinates). For each augmented view's detections $\hat{\mathcal{B}}_m$, we solve the matching problem using the Hungarian algorithm (`linear_sum_assignment`). The cost matrix is defined as $C_{ij} = 1 - \text{IoU}(b_i^{ref}, \hat{b}_{j,m})$. A match is considered valid only if $\text{IoU} \geq \tau$, with $\tau = 0.1$. This reference-based matching ensures that candidates must be spatially consistent with the anchor $\mathcal{B}_{ref}$, preventing spurious detections from being consolidated.

**Quantifying Reliability ($\sigma$).** For each anchor box $b_j \in \mathcal{B}_{ref}$, we compute its reliability $\sigma_j$ by fusing two signals:

1. **Consensus ($C_{ns}$):** Calculated as $\frac{1 + N_{matched}}{M + 1}$, where $N_{matched}$ is the number of augmented views (out of $M = 7$) whose detections successfully matched anchor box $b_j$ ($\text{IoU} \geq 0.1$).

2. **Consistency ($C_{st}$):** The average IoU of successful matches: $\frac{1}{N_{matched}} \sum_m \text{IoU}(b_j^{ref}, \hat{b}_{j,m})$ where the sum is over matched detections.

Final reliability is $\sigma_j = \omega_1 C_{ns} + \omega_2 C_{st}$, with $\omega_1 = 0.6, \omega_2 = 0.4$ as defined in Sec. C.4.

### C.4. Hyperparameter

*Table 9.* DDL Hyperparameters.

| Component | Parameter | Value |
|-----------|-----------|-------|
| Inference | Hardware Infrastructure | NVIDIA H800 GPUs |
|           | Computing Precision | `bfloat16` |
|           | GPU Memory Utilization | 0.98 |
|           | VLLM Batch Size | 32 |
|           | Context Window (`max_tokens`) | 4096 |
|           | Max Generation Tokens | 1024 |
| V-PUP     | Stochastic Samples ($M$) | 7 (Total 8 views with reference) |
| RHC       | Match Threshold ($\tau$) | 0.1 (IoU) |
|           | Consensus Weight ($\omega_1$) | 0.6 |
|           | Consistency Weight ($\omega_2$) | 0.4 |
| DAPE      | Optimizer Configuration | Gemini 3 Flash (Preview) (*OpenRouter*) |

## D. Experimental Details

In this section, we provide supplementary details regarding the entire systems part of our method.

### D.1. DAPE: Distribution shift analysis

In Figure 3(c), we utilize Kernel Density Estimation (KDE) to visualize the semantic evolution of clinical instructions throughout the DAPE process. To verify the global shift of the instructional manifold, we estimate the probability density function $\hat{f}(y)$ based on the cumulative population history $\mathcal{H}^{(g)}$ at durint the iterations. The density estimator for a performance score $y$ is defined as:

$$\hat{f}(y) = \frac{1}{nh} \sum_{i=1}^{n} K\left(\frac{y - y_i}{h}\right), \quad K(u) = \frac{1}{\sqrt{2\pi}} e^{-\frac{1}{2} u^2} \qquad (6)$$

where $n$ is the population size at generation $g$. The smoothing bandwidth $h$ is automatically selected using Scott's Rule ($h = n^{-1/(d+4)}$ for $d = 1$), ensuring an optimal balance between density smoothness and sample variance. To dissect how the meta-optimizer $\Psi$ distinguishes success factors from failures, we perform a *Median Split* on $\mathcal{H}^{(g)}$ at each visualization step. This partitions the cumulative pool into two segments: the "Good" pool (Top 50%) and the "Bad" pool (Bottom 50%). As illustrated in Figure 3(c), the progressive rightward shift of the peak densities and the sharpening of the "Good" distribution justify that DAPE effectively re-shapes the instructional manifold toward high-performance regions rather than merely identifying stochastic outliers.

### D.2. Linguistic vs. Visual Uncertainty

We evaluate the operational differences in uncertainty between linguistic and visual modalities.

**Linguistic Uncertainty:** To estimate the stochasticity of the linguistic decoder, we generate $M = 8$ independent responses for each query using high-temperature sampling ($T = 1.0$). These experiments are conducted locally using the `vLLM` engine to ensure reproducible throughput. The spatial variability across these 8 samples serves as the measure of linguistic uncertainty.

**Visual Uncertainty:** Following the V-PUP protocol, we generate 7 stochastic perturbations utilizing spatial-preserving transformations (rotations within $\pm 3°$, scaling, translations, and horizontal flips). Together with the original image, these constitute a total evidence pool of $M = 8$ views. For this specific comparison, we employ a *simple average* (SA) of the hypothesis coordinates to isolate the effects of the input modality from the consolidation algorithm.

### D.3. RHC: Structured Spatial Consensus

We provide a comparative analysis of different consensus strategies, categorizing them into reference-free heuristics and structured assignment:

- **Simple Average (SA):** A reference-free baseline that aggregates all $M$ bounding box hypotheses from multiple augmented views. The final prediction is generated by computing the arithmetic mean of the $[x_{\min}, y_{\min}, x_{\max}, y_{\max}]$ coordinates after applying inverse spatial transformations.

- **Weighted Average (WA):** An extension of SA where hypotheses are aligned to the original model output via the Hungarian algorithm. Each hypothesis is weighted by its spatial overlap (IoU) with the original prediction, prioritizing views that maintain high spatial consistency with the primary inference.

- **DBSCAN Clustering:** A density-based approach that groups hypotheses using $1 - \text{IoU}$ as the distance metric. Unlike SA, this strategy employs *median* aggregation within each cluster to suppress the influence of heavy-tailed outliers and generates confidence scores based on cluster density and internal spatial tightness.

- **Robust Hungarian Consensus (RHC, Ours):** Our proposed method treats the original output as a canonical anchor. We formulate a global bipartite matching problem to align view-specific hypotheses to this anchor. The final confidence is derived from a multi-faceted consensus score, integrating the *alignment frequency* (how many views agree) and *spatial stability* (mean IoU across matches).

### D.4. Scaling experiments

In Table 3, we report results across different model scales (3B, 7B, 32B, 72B). To ensure statistical significance, all reported metrics (mAP@25, 50, 75) are the average of 3 independent runs with different random seeds. The "Vanilla" baseline represents the zero-shot performance of the base Qwen2.5-VL models using a standard prompt. For a fair comparison, all inference parameters—including a fixed batch size and greedy decoding ($T = 0$) for baseline evaluations—are kept consistent across all model sizes and benchmarks.

### D.5. Confidence calibration

In this section, we provide the computational details for the self-awareness and calibration analysis presented in Sec. 4.2.

**Metric Definitions (Table 4):** To quantify the alignment between the model's internal reliability score $\sigma$ (the normalized consensus score derived from RHC) and its actual grounding performance, we compute the following statistical measures:

1. **Correlation Coefficients:** We primarily utilize the Pearson correlation coefficient ($r$) to measure the linear relationship between $\sigma$ and the Ground Truth IoU (GT-IoU). To ensure robustness, we also compute Spearman's $\rho$ and Kendall's $\tau$ to capture non-linear monotonic relationships.

2. **Statistical Significance:** For all correlation measures, we report $p$-values with the following markers: *** ($p < 0.001$), ** ($p < 0.01$), * ($p < 0.05$), and $ns$ ($p \geq 0.05$).

3. **Mean Absolute Error (MAE):** Calculated as $\frac{1}{N} \sum_{i=1}^{N} |\sigma_i - \text{IoU}_i|$, representing the global calibration gap between perceived reliability and actual precision.

4. **Confidence Dispersion ($\sigma_{conf}$):** The standard deviation of the reliability scores, used to identify if a model's confidence distribution is collapsing (blindly confident) or appropriately sparse (honest uncertainty).

**Reliability Diagram Construction (Figure 5):** The visualization of calibration dynamics follows a systematic procedure:

1. **Binning:** The predicted confidence range $[0, 1]$ is partitioned into $N_{bins} = 10$ equally spaced intervals.

2. **Averaging:** For each bin, we calculate the mean predicted confidence and the mean actual IoU.

3. **95% Confidence Intervals (CI):** We estimate bin uncertainty using $1.96 \cdot \frac{s_j}{\sqrt{n_j}}$, where $s_j$ is the standard deviation and $n_j$ is the sample count in the bin.

### D.6. Baseline implementation details

In Table 5 and Table 6, we provide a detailed description of the configurations for baseline methods to ensure reproducibility.

**Manual Prompting Baselines:** These methods utilize the frozen backbone model with various prompting strategies. Unless otherwise specified, we adhere to the original configurations described in the respective literature:

- **CoT (Chain-of-Thought):** Guides models to break down reasoning into sequential steps following (Wei et al., 2022).

- **Visual Description:** Prompts the model to generate a structured linguistic description of visual features before localization following (Menon & Carl, 2023).

- **Role-Playing:** Prefaces instructions with a senior neuroradiologist persona following (Shanahan et al., 2023).

- **Strict Constraints:** Implements rigorous spatial and formatting constraints within the prompt following (Zhu et al., 2023).

- **VISER:** We follow the protocol in (Izadi et al., 2025) to implement three-way partitioning and contribution analysis.

- **Visual-Instruction Prompting:** Following (Wang et al., 2025), we inject the task instructions directly into the image-level conditioning, enforcing semantic constraints within the visual manifold.

**Automated Optimizer Baselines:** These baselines automate the search for high-performance instructions using the same optimizer backbone as ours, with Gemini 3 Flash (Preview) serving as the meta-optimizer.

- **Meta-Optimization (Meta Opt.):** Implements the iterative optimization-by-prompting framework following (Yang et al., 2023).

- **Gradient-based Optimization (Gradient Opt.):** Implements the textual gradient descent approach described in (Pryzant et al., 2023), adapted for the brain MRI domain.

**Training-based Baselines:** For fine-tuning baselines, we utilize the `LLaMA-Factory` framework (Zheng et al., 2024) with the following key parameters:

- **LoRA (Low-Rank Adaptation):** Applied to all linear layers with rank $r = 8$ and $r = 16$. The training uses a learning rate of $1 \times 10^{-6}$, a cutoff length of 4096.

- **Full SFT (Supervised Fine-Tuning):** We perform full-parameter fine-tuning with a learning rate of $1 \times 10^{-6}$, a cutoff length of 4096.

# E. Additional Results

## E.1. Comprehensive baseline comparisons

Table 10 presents the full clinical grounding results on both BTD and NOVA benchmarks. DDL consistently outperforms individual prompting strategies and automated optimizers, particularly at the high-precision mAP@75 threshold. Due to GPU memory constraints and infrastructure limitations, training-based baselines (SFT/LoRA) were not performed for the 72B model.

## E.2. Detailed calibration analysis across scales

We investigate the scaling effect on model self-awareness by analyzing the correlation between the consensus reliability score $\sigma$ and actual grounding performance. Figures 7 and 8 provide the quantitative assessment for both benchmarks.

## E.3. Divergence of visual and linguistic uncertainty

We further expand on the results regarding the mismatch between visual and linguistic stability. As shown in Table 11, linguistic uncertainty (generated via $T = 1.0$ sampling) often represents stochastic noise in the decoder's token selection.

## E.4. Additional qualitative visualizations

We provide additional grounding trajectories illustrating how DAPE refines the Focus (Stage 1) and how RHC stabilizes the final coordinates (DDL) across various rare pathologies, including glioblastoma variants and micro-calcifications. Figure 9 visualizes the iterative improvements on the development set, while Figures 10–13 show specific evolutionary samples across model scales. The final grounding visualizations of our DDL outputs are shown in Figures 14–17.

## E.5. Sensitivity and ablation analyses

We further provide sensitivity and ablation analyses in Table 12. Panel A studies the accuracy-efficiency trade-off as the number of visual views increases, showing that performance improves with additional views while the view-wise computation remains parallelizable. Panel B isolates the contributions of DAPE and the full DDL pipeline, indicating that prompt optimization already improves the vanilla baseline and that visual verification provides additional gains. Panel C evaluates sensitivity to the optimizer LLM, while Panel D tests whether optimized prompts transfer across target backbones. These results suggest that DAPE is not tied to a single proprietary optimizer or a single target model. Panel E examines reliability-score thresholding for hallucination suppression, and Panel F reports Wilcoxon signed-rank tests, confirming that the mAP@50 gains are statistically significant across datasets and model scales.

*Table 10.* Comprehensive performance comparison of DDL against training-free and training-based baselines across multiple model scales on the BTD and NOVA benchmarks.

| Size | Method | BTD | | | NOVA | | |
|------|--------|---------|---------|---------|---------|---------|---------|
| | | mAP@25 | mAP@50 | mAP@75 | mAP@25 | mAP@50 | mAP@75 |
| | Vanilla | 0.418 | 0.370 | 0.269 | 0.221 | 0.088 | 0.040 |
| | COT | 0.256 | 0.171 | 0.055 | 0.092 | 0.019 | 0.003 |
| | Visual-Des. | 0.340 | 0.289 | 0.220 | 0.197 | 0.067 | 0.028 |
| | Role-Play | 0.355 | 0.304 | 0.230 | 0.226 | 0.094 | 0.038 |
| | Strict Const. | 0.337 | 0.297 | 0.209 | 0.244 | 0.113 | 0.049 |
| 3B | VISER | 0.412 | 0.365 | 0.257 | 0.239 | 0.100 | 0.038 |
| | Visual-Instruc. | 0.191 | 0.123 | 0.083 | 0.036 | 0.010 | 0.002 |
| | Gradient Optim. | 0.188 | 0.180 | 0.124 | 0.039 | 0.009 | 0.003 |
| | Meta Optim. | 0.408 | 0.357 | 0.269 | 0.282 | 0.121 | 0.049 |
| | LoRA-R8 | 0.429 | 0.379 | 0.277 | 0.225 | 0.092 | 0.042 |
| | SFT | **0.479** | **0.413** | **0.308** | 0.283 | 0.128 | 0.046 |
| | **Ours** | 0.403 | 0.379 | 0.283 | **0.298** | **0.150** | **0.066** |
| | Vanilla | 0.348 | 0.289 | 0.154 | 0.286 | 0.135 | 0.036 |
| | COT | 0.110 | 0.082 | 0.027 | 0.100 | 0.052 | 0.011 |
| | Visual-Des. | 0.217 | 0.181 | 0.101 | 0.201 | 0.084 | 0.025 |
| | Role-Play | 0.316 | 0.255 | 0.125 | 0.292 | 0.140 | 0.038 |
| | Strict Const. | 0.270 | 0.207 | 0.085 | 0.237 | 0.113 | 0.031 |
| 7B | VISER | 0.225 | 0.139 | 0.076 | 0.193 | 0.063 | 0.017 |
| | Visual-Instruc. | 0.164 | 0.140 | 0.058 | 0.188 | 0.086 | 0.022 |
| | Gradient Optim. | 0.130 | 0.081 | 0.023 | 0.125 | 0.046 | 0.015 |
| | Meta Optim. | 0.249 | 0.221 | 0.134 | 0.327 | 0.147 | 0.039 |
| | LoRA-R8 | 0.345 | 0.289 | 0.148 | 0.287 | 0.135 | 0.036 |
| | SFT | **0.578** | **0.462** | **0.263** | 0.360 | 0.182 | 0.044 |
| | **Ours** | 0.383 | 0.302 | 0.159 | **0.369** | **0.206** | **0.075** |
| | Vanilla | 0.496 | 0.367 | 0.160 | 0.406 | 0.208 | 0.058 |
| | COT | 0.328 | 0.166 | 0.047 | 0.182 | 0.108 | 0.023 |
| | Visual-Des. | 0.524 | 0.396 | 0.163 | 0.403 | 0.218 | 0.053 |
| | Role-Play | 0.297 | 0.159 | 0.046 | 0.150 | 0.092 | 0.027 |
| | Strict Const. | 0.497 | 0.377 | 0.159 | 0.348 | 0.190 | 0.049 |
| 32B | VISER | 0.406 | 0.291 | 0.083 | 0.290 | 0.131 | 0.027 |
| | Visual-Instruc. | 0.210 | 0.114 | 0.028 | 0.154 | 0.053 | 0.010 |
| | Gradient Optim. | 0.188 | 0.070 | 0.001 | 0.087 | 0.017 | 0.003 |
| | Meta Optim. | 0.531 | 0.374 | 0.151 | 0.371 | 0.202 | 0.056 |
| | LoRA-R8 | 0.494 | 0.368 | 0.166 | 0.395 | 0.201 | 0.050 |
| | LoRA-R16 | 0.490 | 0.371 | 0.165 | 0.399 | 0.206 | 0.051 |
| | **Ours** | **0.602** | **0.433** | **0.206** | **0.454** | **0.266** | **0.096** |
| | Vanilla | 0.571 | 0.488 | 0.306 | 0.411 | 0.245 | 0.065 |
| | COT | 0.463 | 0.368 | 0.120 | 0.270 | 0.168 | 0.052 |
| | Visual-Des. | 0.606 | 0.510 | 0.305 | 0.343 | 0.200 | 0.051 |
| | Role-Play | 0.625 | 0.520 | 0.297 | 0.466 | 0.268 | 0.063 |
| 72B | Strict Const. | 0.466 | 0.417 | 0.245 | 0.204 | 0.122 | 0.033 |
| | VISER | 0.543 | 0.470 | 0.235 | 0.324 | 0.152 | 0.031 |
| | Visual-Instruc. | **0.701** | 0.568 | 0.338 | 0.361 | 0.210 | 0.058 |
| | Meta Optim. | 0.603 | 0.526 | 0.287 | 0.471 | 0.269 | 0.064 |
| | **Ours** | 0.650 | **0.572** | **0.346** | **0.500** | **0.301** | **0.107** |

*Table 11.* Comparison of linguistic and visual uncertainty quantification across model sizes (3B to 72B) and prompt configurations. Performance is evaluated across difficulty percentiles (@25, @50, @75) using Softmax Averaging (SA) and Weighted Averaging (WA). The Δ column reflects the relative shift in uncertainty metrics when incorporating visual information. **Takeaway: visual uncertainty provides more reliable grounding signals than language, and optimal prompts exhibit more stable performance.**

| P. | Size | M. | Language | | | Visual | | | Δ (%) (Visual.→Language.) | | |
|---|---|---|---|---|---|---|---|---|---|---|---|
| | | | @25 | @50 | @75 | @25 | @50 | @75 | @25 | @50 | @75 |
| **Optimal Prompt** | 3B | SA | 0.274 | 0.127 | 0.052 | 0.319 | 0.122 | 0.042 | ▲16.6% | ▼4.5% | ▼19.5% |
| | | WA | 0.279 | 0.131 | 0.052 | 0.292 | 0.148 | 0.057 | ▲4.4% | ▲12.9% | ▲10.8% |
| | 7B | SA | 0.321 | 0.141 | 0.038 | 0.370 | 0.160 | 0.039 | ▲15.2% | ▲13.7% | ▲2.9% |
| | | WA | 0.301 | 0.135 | 0.035 | 0.360 | 0.187 | 0.063 | ▲19.7% | ▲38.0% | ▲81.9% |
| | 32B | SA | 0.416 | 0.214 | 0.049 | 0.445 | 0.218 | 0.065 | ▲7.1% | ▲1.9% | ▲32.6% |
| | | WA | 0.412 | 0.224 | 0.047 | 0.445 | 0.208 | 0.065 | ▲8.2% | ▼7.2% | ▲38.0% |
| | 72B | SA | 0.457 | 0.260 | 0.060 | 0.487 | 0.273 | 0.078 | ▲6.7% | ▲4.8% | ▲30.2% |
| | | WA | 0.459 | 0.269 | 0.062 | 0.493 | 0.303 | 0.101 | ▲7.5% | ▲11.8% | ▲71.8% |
| **Vanilla Prompt** | 3B | SA | 0.222 | 0.089 | 0.041 | 0.319 | 0.122 | 0.042 | ▲16.4% | ▲0.0% | ▼30.6% |
| | | WA | 0.221 | 0.088 | 0.043 | 0.292 | 0.148 | 0.057 | ▲3.1% | ▲15.4% | ▼2.6% |
| | 7B | SA | 0.293 | 0.128 | 0.034 | 0.370 | 0.160 | 0.039 | ▲3.5% | ▼1.7% | ▼19.9% |
| | | WA | 0.298 | 0.135 | 0.036 | 0.360 | 0.187 | 0.063 | ▼2.2% | ▲21.0% | ▲46.8% |
| | 32B | SA | 0.409 | 0.204 | 0.052 | 0.445 | 0.218 | 0.065 | ▲8.3% | ▼3.9% | ▲2.3% |
| | | WA | 0.408 | 0.211 | 0.059 | 0.445 | 0.208 | 0.065 | ▲5.3% | ▲18.7% | ▲34.2% |
| | 72B | SA | 0.429 | 0.229 | 0.058 | 0.487 | 0.273 | 0.078 | ▲4.5% | ▲2.0% | ▲17.6% |
| | | WA | 0.402 | 0.215 | 0.055 | 0.493 | 0.303 | 0.101 | ▲13.6% | ▲37.3% | ▲71.5% |

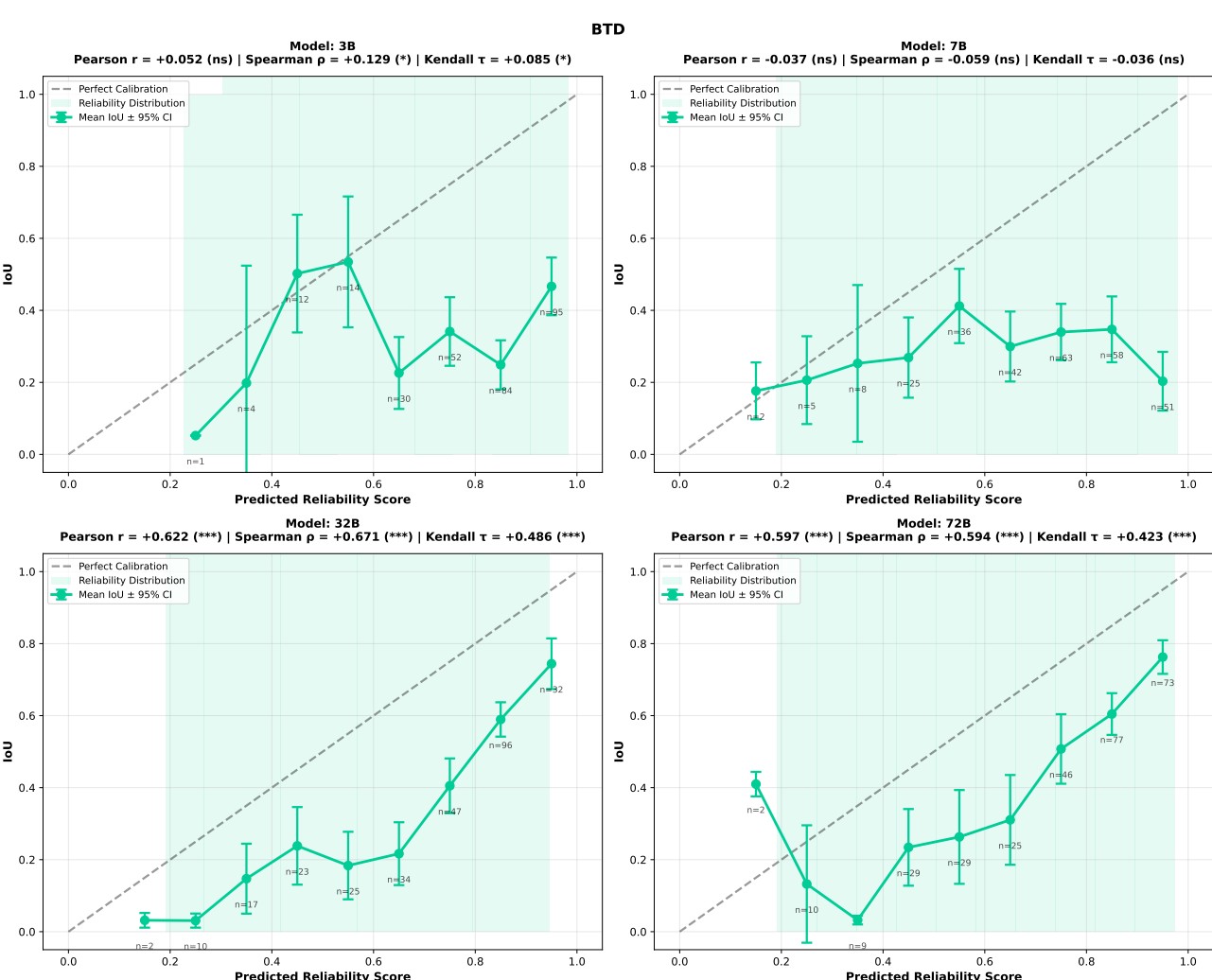

*Figure 7.* Calibration analysis across model scales on the BTD dataset.

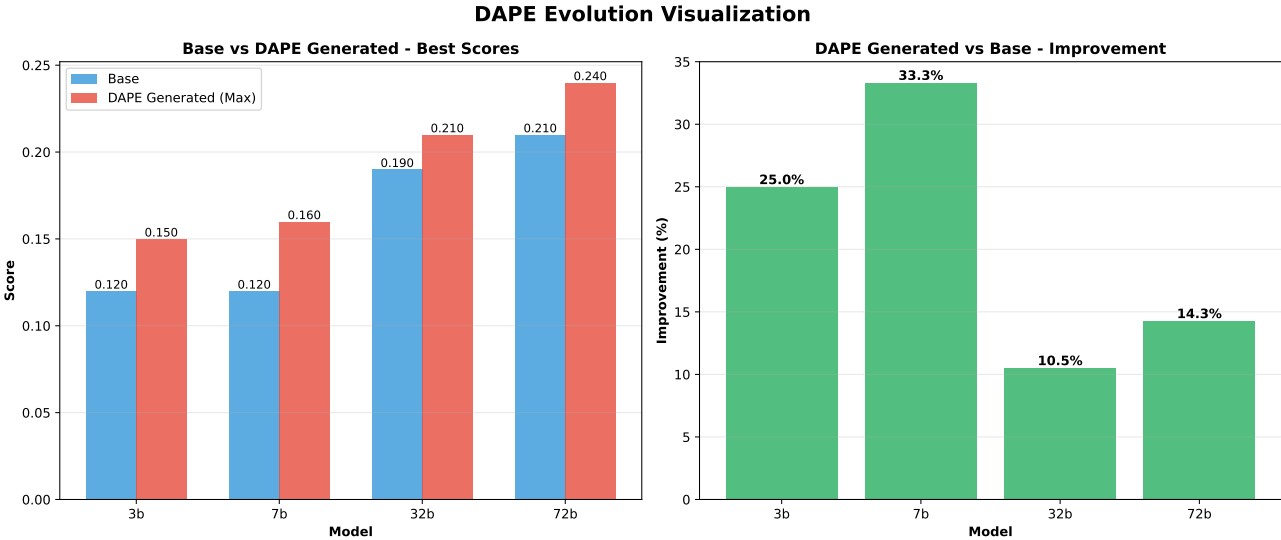

*Figure 8.* Calibration analysis across model scales on the NOVA dataset.

*Figure 9.* Visualization of grounding improvement driven by DAPE iterations on the development set.

**Highlight Legend:**
Green = Words added in current generation
Red = Words removed from previous generation

## qwen2.5-vl-3b

### Best Prompt (Best DAPE)

```
Identify any visually anomalous regions in this medical scan. Focus on detecting areas with abnormal signal intensity (atypical
brightness or darkness), irregular textures, or mass-like structures that deviate from the surrounding healthy tissue.

Return the bounding boxes of these abnormal areas in JSON format:
If no target is detected, return the string: "no target".
If detected, return a list of 2D bounding boxes in the following JSON format:
```json
[
  {"bbox_2d": [x1, y1, x2, y2], "label": "visual_anomaly"}
]
```
```

| Prompt Name | Source | Score | Gen | Prompt Text |
|---|---|---|---|---|
| **base** | Base | **0.120** | 0 | Return bounding boxes of any abnormal areas as JSON format:
If the image do not have the target, return the string: "no target".
If detected, return a list of 2D bounding boxes around the target regions in the following JSON format:
```json
[
{"bbox_2d": [x1, y1, x2, y2], "label": "label"},
...
]
``` |
| **DAPE_gen_0** | DAPE Generated | **0.070** | 0 | Identify and localize any visually anomalous regions in this medical scan. Focus on detecting:
- Variations in signal intensity (atypical hyperintense or hypointense areas).
- Structural distortions, mass effects, or irregular tissue boundaries.
- Asymmetric growths or focal lesions.

Return the findings as a JSON list of 2D bounding boxes. If no abnormalities are present, return "no target".

```json
[
{"bbox_2d": [x1, y1, x2, y2], "label": "abnormality"}
]
``` |
| **DAPE_gen_1** | DAPE Generated | **0.140** | 1 | Identify ~~and localize~~ any visually anomalous regions in this medical scan. Focus on ~~detecting:~~ **detecting** – ~~Variations in~~ **areas with abnormal** signal intensity (atypical ~~hyperintense~~ **brightness** or ~~darkness),~~ **darkness), irregular textures,** ~~hypointense areas).~~ – ~~Structural distortions,~~ **mass** ~~effects,~~ or ~~irregular~~ **mass-like structures that deviate from the surrounding healthy** ~~tissue~~ **tissue.** ~~boundaries.~~ – ~~Asymmetric growths or focal lesions.~~ Return the ~~findings~~ **as bounding boxes of these abnormal areas in JSON format: If no target is detected, return the string: "no target". If detected, return** a ~~JSON~~ list of 2D bounding ~~boxes.~~ **boxes in the following JSON format:** ~~If no abnormalities are present, return "no target".~~ ```json [ {"bbox_2d": [x1, y1, x2, y2], "label": ~~"abnormality"}~~ **"visual_anomaly"}** ] ``` |
| **DAPE_gen_2** | DAPE Generated | **0.120** | 2 | Identify any visually anomalous regions in this medical scan. Focus on detecting areas with abnormal signal intensity (atypical brightness or darkness), irregular textures, or mass-like structures that deviate from the surrounding healthy tissue. Return the bounding boxes of these **visually** abnormal areas in JSON format: If no target is detected, return the string: "no target". If detected, return a list of 2D bounding boxes in the following JSON format: ```json [ {"bbox_2d": [x1, y1, x2, y2], "label": "visual_anomaly"} ] ``` |
| **DAPE_gen_3** | DAPE Generated | **0.130** | 3 | Identify any visually anomalous regions in this medical scan. Focus on detecting areas with abnormal signal intensity (atypical brightness or darkness), irregular textures, or mass-like structures that deviate from the surrounding healthy tissue. Return the bounding boxes of these visually abnormal areas in JSON format: If no target is detected, return the string: "no target". If detected, return a list of 2D bounding boxes in the following JSON format: ```json [ {"bbox_2d": [x1, y1, x2, y2], "label": "visual_anomaly"} ] ``` |

*Figure 10.* Evolutionary trajectory of DAPE instruction samples for the Qwen2.5-VL-3B model.

**Highlight Legend:**
Green = Words added in current generation
Red = Words removed from previous generation

## qwen2.5-vl-7b

### Best Prompt (Best DAPE)

```
Detect and localize any visually anomalous regions in this medical scan. Focus on identifying areas that exhibit:
- Focal signal intensity changes (abnormally hyperintense or hypointense regions) relative to surrounding tissue.
- Structural irregularities, mass-like effects, or disrupted anatomical boundaries.
- Distinct textural changes or non-symmetric focal lesions.

Return the bounding boxes of these visually abnormal areas in JSON format:
If the image does not have the target, return the string: "no target".
If detected, return a list of 2D bounding boxes in the following JSON format:
```json
[
{"bbox_2d": [x1, y1, x2, y2], "label": "visual_anomaly"}
]
```
```

| Prompt Name | Source | Score | Gen | Prompt Text |
|---|---|---|---|---|
| **base** | Base | **0.120** | 0 | Return bounding boxes of any abnormal areas as JSON format: If the image do not have the target, return the string: "no target". If detected, return a list of 2D bounding boxes around the target regions in the following JSON format: ```json [ {"bbox_2d": [x1, y1, x2, y2], "label": "label"}, ... ] ``` |
| **DAPE_gen_0** | DAPE Generated | **0.130** | 0 | Identify and localize any visually anomalous regions in this medical scan. Focus on detecting: - Focal signal intensity changes (areas significantly brighter or darker than surrounding tissue). - Distinct mass-like structures or irregular tissue boundaries. - Disruptions in expected anatomical patterns. Return the bounding boxes of these abnormal areas in JSON format. If no abnormalities are detected, return the string: "no target". If detected, return a list of 2D bounding boxes in the following JSON format: ```json [ {"bbox_2d": [x1, y1, x2, y2], "label": "visual_anomaly"} ] ``` |
| **DAPE_gen_1** | DAPE Generated | **0.130** | 1 | **Detect** ~~Identify and localize~~ any visually anomalous regions in this medical scan. Focus on ~~detecting: identifying areas with:~~ - ~~Focal~~ **Abnormal** signal intensity **(hyperintense** ~~changes (areas significantly brighter~~ or ~~darker than~~ **hypointense regions) relative to** surrounding ~~tissue).~~ **tissue.** - ~~Distinct mass-like structures~~ **Structural irregularities, mass effects,** or ~~irregular tissue~~ **disrupted anatomical** boundaries. - ~~Disruptions in~~ **Non-symmetric or** ~~expected~~ **unexpected** ~~anatomical patterns.~~ **focal lesions.** Return the bounding boxes of these abnormal areas in JSON ~~format.~~ **format:** If **the image does** ~~no~~ **not** ~~abnormalities are detected,~~ **have the target,** return the string: "no target". If detected, return a list of 2D bounding boxes in the following JSON format: ```json [ {"bbox_2d": ~~[x1, y1,~~ **[y1,** ~~x2, x1,~~ **x1,** ~~y2],~~ **y2, x2],** "label": ~~"visual_anomaly"}~~ **"abnormality"}** ] ``` |
| **DAPE_gen_2** | DAPE Generated | **0.110** | 2 | ~~Detect~~ **Identify and localize** any visually anomalous regions in this medical scan. Focus on ~~identifying~~ **detecting** areas ~~with:~~ **that deviate from standard anatomical patterns, specifically:** - ~~Abnormal~~ **Focal** signal intensity ~~(hyperintense~~ **changes (regions significantly brighter** or darker ~~than~~ **than surrounding** ~~hypointense regions) relative to~~ ~~tissue.~~ **tissue).** - **Mass-like structures** ~~Structural irregularities, mass effects,~~ or ~~disrupted anatomical~~ **irregular tissue** boundaries. - ~~Non-symmetric~~ **Disruptions in the expected** ~~or unexpected focal lesions.~~ **symmetry** **texture of the anatomy.** Return the bounding boxes of these **visually** abnormal areas in JSON format: If ~~the image does~~ ~~not have the target,~~ **abnormalities are detected,** return the string: "no target". If detected, return a list of 2D bounding boxes in the following JSON format: ```json [ {"bbox_2d": ~~[y1, x1,~~ **[x1, y2,** ~~y1, x2],~~ **x2, y2],** "label": ~~"abnormality"}~~ **"visual_anomaly"}** ] ``` |
| **DAPE_gen_3** | DAPE Generated | **0.110** | 3 | Identify and localize any visually anomalous regions in this medical scan. Focus on ~~detecting~~ **detecting:** ~~areas that deviate from standard anatomical patterns, specifically:~~ - ~~Focal~~ **Areas of abnormal** signal intensity ~~changes (regions significantly~~ **(significantly** brighter or darker than surrounding tissue). - **Irregular tissue textures,** ~~Mass-like~~ **mass-like** ~~structures~~ **structures,** or ~~irregular tissue boundaries.~~ **disruptions in anatomical patterns.** - ~~Disruptions in the expected~~ **Distinct** boundaries that deviate from healthy **structural** ~~symmetry~~ **symmetry.** ~~or texture of the anatomy.~~ Return the bounding boxes of these visually abnormal areas in JSON format: If no abnormalities are detected, return the string: "no target". If detected, return a list of 2D bounding boxes in the following JSON format: ```json [ {"bbox_2d": [x1, y1, x2, y2], "label": "visual_anomaly"} ] ``` |

*Figure 11.* Evolutionary trajectory of DAPE instruction samples for the Qwen2.5-VL-7B model.

**Highlight Legend:**
Green = Words added in current generation
Red = Words removed from previous generation

## qwen2.5-vl-32b

### Best Prompt (Best DAPE)

```
Scan this MRI image for any visually abnormal regions or pathological findings. Focus on identifying:
- Signal intensity changes (unusual brightness or darkness).
- Mass effects, irregular textures, or structural distortions.
- Lesions, tumors, or areas of edema.

Return the results strictly in the following JSON format. If no abnormality is detected, return: "no target".

```json
[
{"bbox_2d": [x1, y1, x2, y2], "label": "abnormal_finding"}
]
```
```

| Prompt Name | Source | Score | Gen | Prompt Text |
|---|---|---|---|---|
| **base** | Base | **0.190** | 0 | Return bounding boxes of any abnormal areas as JSON format: If the image do not have the target, return the string: "no target". If detected, return a list of 2D bounding boxes around the target regions in the following JSON format: ```json [ {"bbox_2d": [x1, y1, x2, y2], "label": "label"}, ... ]``` |
| **DAPE_gen_0** | DAPE Generated | **0.178** | 0 | Examine the MRI scan for pathological abnormalities. Pay close attention to tissue texture, structural integrity, and signal intensity. Scan for: 1. Space-occupying lesions (tumors, cysts). 2. Areas of inflammation or edema. 3. Structural anomalies or vascular abnormalities. Return the result strictly in the following JSON format. If no pathology is detected, return: "no target". ```json [ {"bbox_2d": [x1, y1, x2, y2], "label": "pathology_type"} ]``` |
| **DAPE_gen_1** | DAPE Generated | **0.177** | 1 | Examine the MRI scan for pathological abnormalities. ~~Pay close attention to~~ **Focus on detecting changes in** tissue ~~texture, density,~~ structural ~~integrity,~~ **displacement,** and signal intensity. Scan for: 1. ~~Space-occupying lesions (tumors,~~ **Tumors,** ~~cysts).~~ **cysts, or masses.** 2. **Edema** ~~Areas of inflammation~~ or ~~edema.~~ **inflammatory changes.** 3. Structural anomalies or **lesions.** ~~vascular abnormalities.~~ Return the result strictly in the following JSON format. If no ~~pathology~~ **target** is detected, return: "no target". ```json [ {"bbox_2d": [x1, y1, x2, y2], "label": "pathology_type"} ]``` |
| **DAPE_gen_2** | DAPE Generated | **0.200** | 2 | ~~Examine the~~ **Scan this** MRI ~~scan~~ **image** for **any visually abnormal regions or** pathological ~~abnormalities.~~ **findings.** Focus on ~~detecting~~ **identifying:** - **Signal intensity** changes ~~in tissue density,~~ **(unusual brightness or darkness). - Mass effects, irregular textures, or** structural **distortions. - Lesions,** ~~displacement, and signal intensity. Scan for: 1. Tumors, cysts, or areas of edema.~~ ~~masses. 2. Edema or inflammatory changes. 3. Structural anomalies or lesions.~~ Return the ~~result~~ **results** strictly in the following JSON format. If no ~~target~~ **abnormality** is detected, return: "no target". ```json [ {"bbox_2d": [x1, y1, x2, y2], "label": ~~"pathology_type"}~~ **"abnormal_finding"}** ]``` |
| **DAPE_gen_3** | DAPE Generated | **0.190** | 3 | Scan this MRI image for any visually abnormal regions or pathological findings. ~~Focus on identifying:~~ **Pay close attention to:** - Signal intensity changes (unusual ~~brightness~~ **hyperintense** or ~~darkness).~~ **hypointense areas).** - Mass effects, irregular textures, or structural distortions. - Lesions, tumors, or areas of ~~edema.~~ **inflammation/edema.** Return the results strictly in the following JSON format. If no abnormality is detected, return: "no target". ```json [ {"bbox_2d": [x1, y1, x2, y2], "label": "abnormal_finding"} ]``` |

*Figure 12.* Evolutionary trajectory of DAPE instruction samples for the Qwen2.5-VL-32B model.

**Highlight Legend:**
Green = Words added in current generation
Red = Words removed from previous generation

## qwen2.5-vl-72b

### Best Prompt (Best DAPE)

Act as an expert radiologist. Carefully examine this medical image to identify and localize any pathological abnormalities, such as tumors, lesions, cysts, or inflammatory changes.

If the image is normal or contains no significant findings, return: "no target".

If abnormalities are detected, provide the precise 2D bounding boxes in the following JSON format:
```json
[
  {"bbox_2d": [x1, y1, x2, y2], "label": "specific_finding_name"}
]
```
Ensure the bounding boxes tightly and accurately encompass the identified pathologies.

| Prompt Name | Source | Score | Gen | Prompt Text |
|---|---|---|---|---|
| **base** | Base | **0.210** | 0 | Return bounding boxes of any abnormal areas as JSON format:
If the image do not have the target, return the string: "no target".
If detected, return a list of 2D bounding boxes around the target regions in the following JSON format:
```json
[
{"bbox_2d": [x1, y1, x2, y2], "label": "label"},
...
]``` |
| **DAPE_gen_0** | DAPE Generated | **0.200** | 0 | Act as an expert radiologist. Carefully examine this medical image to identify and localize any pathological abnormalities, including tumors, lesions, cysts, or inflammatory changes.

If the image is normal or contains no significant findings, return: "no target".

If abnormalities are detected, provide the coordinates as 2D bounding boxes in this JSON format:
```json
[
  {"bbox_2d": [x1, y1, x2, y2], "label": "specific_finding_name"}
]```
Ensure the bounding boxes precisely encompass the identified pathologies. |
| **DAPE_gen_1** | DAPE Generated | **0.200** | 1 | Act as an expert radiologist. Carefully ~~examine~~ **analyze** this medical image to ~~identify~~ **detect** and localize any ~~pathological~~ **clinical** abnormalities, ~~including~~ **such as lesions,** tumors, ~~lesions, cysts,~~ or inflammatory changes. If ~~the image is normal or contains~~ no ~~significant findings,~~ **abnormalities are identified,** return: "no target". If abnormalities are detected, provide the **precise** ~~coordinates as~~ 2D bounding boxes in ~~this~~ **the following** JSON format: ```json [ {"bbox_2d": ~~[x1, y1,~~ **[y1, x2, x1, y2],** **y2, x2],** "label": ~~"specific_finding_name"}~~ **"finding_name"}** ] ``` Ensure the ~~bounding~~ boxes ~~precisely encompass~~ **tightly bound** the identified ~~pathologies.~~ **pathological regions.** |
| **DAPE_gen_2** | DAPE Generated | **0.187** | 2 | Act as an expert radiologist. Carefully ~~analyze~~ **examine** this medical image to ~~detect~~ **identify** and localize any ~~clinical~~ **pathological** abnormalities, such as lesions, tumors, **cysts,** or inflammatory changes. If **the image is normal or contains** no **significant findings,** ~~abnormalities are identified,~~ return: "no target". If abnormalities are detected, provide the ~~precise~~ **coordinates as** 2D bounding boxes in the following JSON format: ```json [ {"bbox_2d": ~~[y1, x1,~~ **[x1, y2, y1, x2],** **x2, y2],** "label": ~~"finding_name"}~~ **"specific_finding_name"}** ] ``` Ensure the **bounding** boxes **precisely and** tightly ~~bound~~ **encompass** the identified ~~pathological~~ **pathologies.** ~~regions.~~ |
| **DAPE_gen_3** | DAPE Generated | **0.230** | 3 | Act as an expert radiologist. Carefully examine this medical image to identify and localize any pathological abnormalities, such as **tumors,** lesions, ~~tumors,~~ cysts, or inflammatory changes. If the image is normal or contains no significant findings, return: "no target". If abnormalities are detected, provide the **precise** ~~coordinates as~~ 2D bounding boxes in the following JSON format: ```json [ {"bbox_2d": [x1, y1, x2, y2], "label": "specific_finding_name"} ] ``` Ensure the bounding boxes ~~precisely~~ **tightly** and ~~tightly~~ **accurately** encompass the identified pathologies. |

*Figure 13.* Evolutionary trajectory of DAPE instruction samples for the Qwen2.5-VL-72B model.

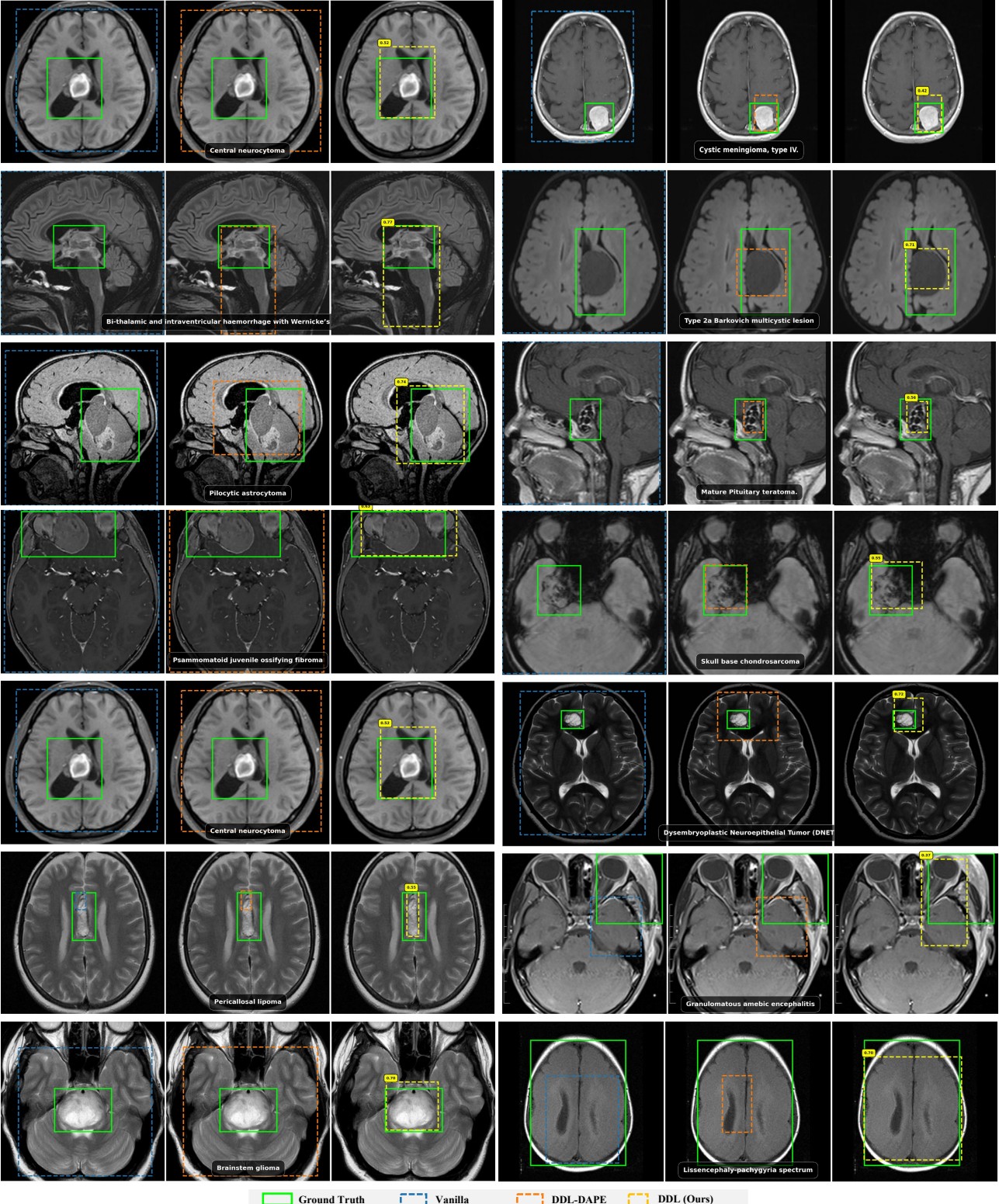

*Figure 14.* Qualitative visualization of the final DDL grounding decisions on the 3B model.

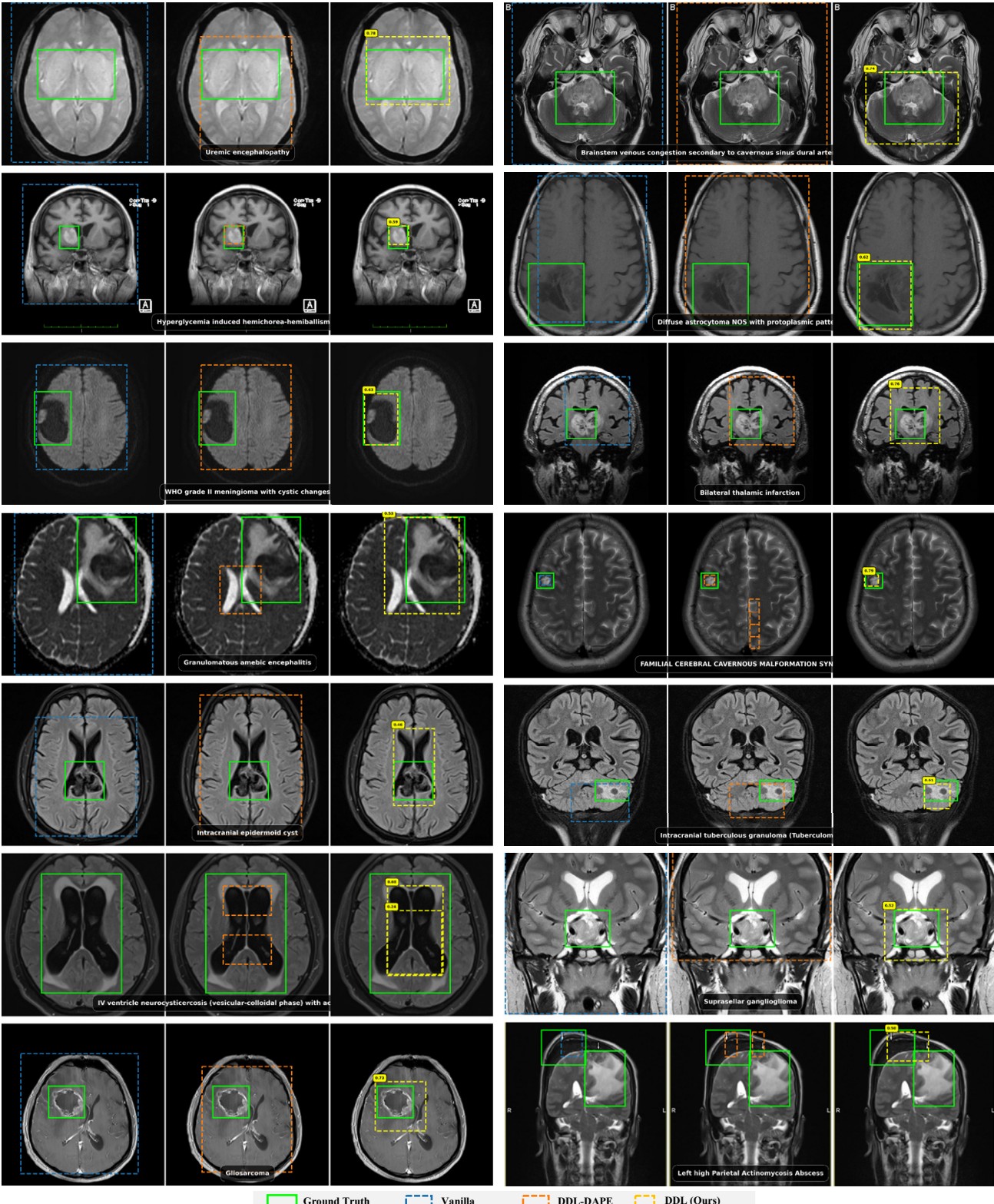

*Figure 15.* Qualitative visualization of the final DDL grounding decisions on the 7B model.

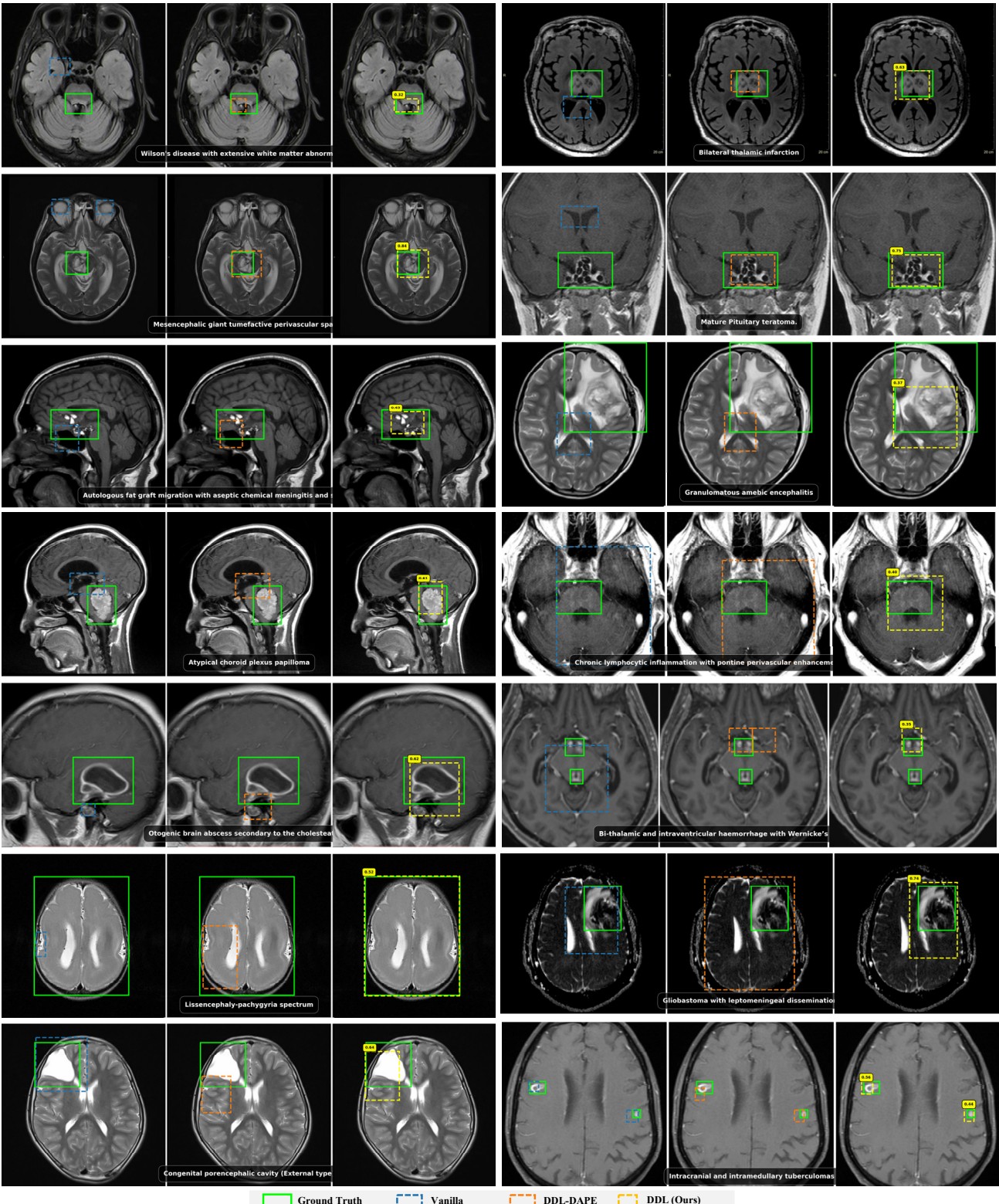

*Figure 16.* Qualitative visualization of the final DDL grounding decisions on the 32B model.

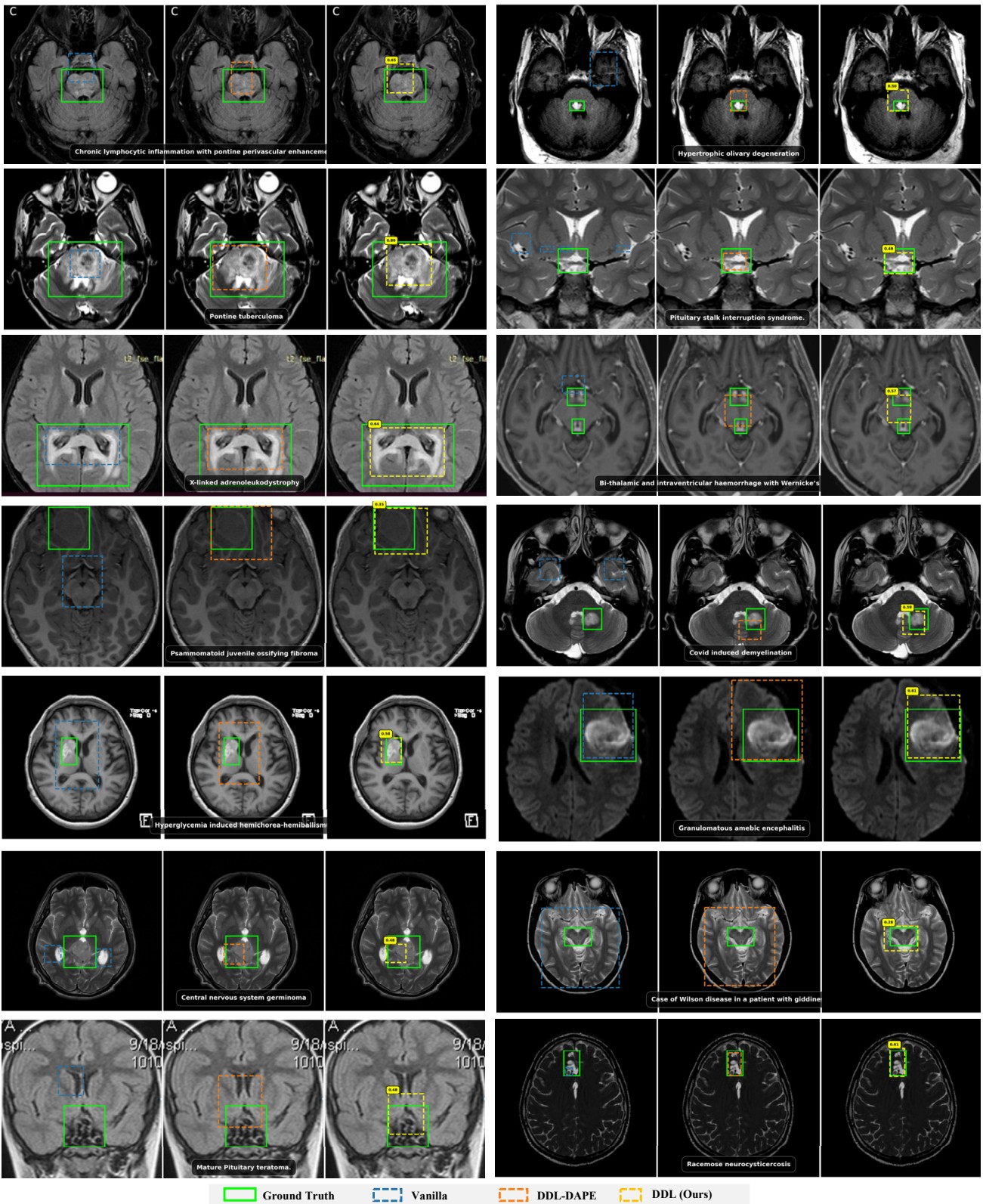

*Figure 17.* Qualitative visualization of the final DDL grounding decisions on the 72B model.

*Table 12.* Sensitivity and ablation analyses of DDL. The table reports mAP@50 under different visual-view counts, component configurations, optimizer LLMs, prompt-transfer settings, CRS thresholds, and statistical tests.

| A. Accuracy-efficiency trade-off across visual views | | | | |
|---|---|---|---|---|
| $M$ | NOVA | BTD | Serial time | Parallel time |
| 1 | 0.094 | 0.370 | 0.19s | 0.19s |
| 3 | 0.122 | 0.378 | 0.49s | ∼0.19s |
| 5 | 0.131 | 0.414 | 0.80s | ∼0.19s |
| 7 | 0.143 | 0.416 | 1.14s | ∼0.19s |

| B. Component ablation in mAP@50 | | | | |
|---|---|---|---|---|
| Dataset | Backbone | Vanilla | +DAPE | DDL |
| NOVA | 3B | 0.088 | 0.132 | 0.150 |
| NOVA | 7B | 0.135 | 0.160 | 0.206 |
| NOVA | 32B | 0.208 | 0.224 | 0.266 |
| NOVA | 72B | 0.245 | 0.270 | 0.301 |
| BTD | 3B | 0.370 | 0.378 | 0.379 |
| BTD | 7B | 0.289 | 0.298 | 0.302 |
| BTD | 32B | 0.367 | 0.422 | 0.433 |
| BTD | 72B | 0.488 | 0.496 | 0.572 |

| C. Optimizer LLM sensitivity on NOVA | | | |
|---|---|---|---|
| Optimizer | Type | 3B | 7B |
| Vanilla | – | 0.088 | 0.135 |
| Qwen 3.5-9B | Open | 0.111 | 0.146 |
| Llama 3.1-8B | Open | 0.122 | 0.147 |
| GPT-4o | Closed | 0.129 | 0.155 |
| Gemini 3.0 Flash | Closed | 0.132 | 0.160 |

| D. Cross-backbone prompt transfer on NOVA | | | |
|---|---|---|---|
| Prompt source | Optimizer | Target | mAP@50 |
| Vanilla | – | 7B | 0.135 |
| 3B-optimized | GPT-4o | 7B | 0.152 |
| 3B-optimized | Llama 3.1-8B | 7B | 0.149 |
| 3B-optimized | Qwen 3.5-9B | 7B | 0.142 |

| E. CRS thresholding on NOVA | | | | |
|---|---|---|---|---|
| Model | Remaining | mAP@50 | Avg. IoU | Hallucination ↓ |
| 3B | 100.0% | 0.150 | 0.231 | 45.1% |
| 3B | 88.9% | 0.177 | 0.241 | 43.4% |
| 3B | 67.5% | 0.185 | 0.249 | 42.9% |
| 7B | 100.0% | 0.206 | 0.254 | 42.0% |
| 7B | 69.1% | 0.272 | 0.308 | 35.1% |
| 7B | 44.2% | 0.310 | 0.339 | 30.3% |

| F. Wilcoxon signed-rank test for DDL vs. vanilla on mAP@50 | | | |
|---|---|---|---|
| Dataset | Backbone | $p$-value | Significance |
| NOVA | 3B | $1.14\times10^{-10}$ | *** |
| NOVA | 7B | $4.55\times10^{-13}$ | *** |
| NOVA | 32B | $1.24\times10^{-3}$ | ** |
| NOVA | 72B | $2.08\times10^{-5}$ | *** |
| BTD | 3B | $4.29\times10^{-11}$ | *** |
| BTD | 7B | $3.71\times10^{-15}$ | *** |
| BTD | 32B | $3.61\times10^{-5}$ | *** |
| BTD | 72B | $9.52\times10^{-12}$ | *** |

*Notes.* Panels A–E report mAP@50 unless otherwise specified. Panel A uses Qwen2.5-VL-3B on 100 fixed random samples per dataset. Hallucinations in Panel E are predictions with IoU below 0.1.

