# OpenReview forum: "Dynamic Decision Learning: Test-Time Evolution for Abnormality Grounding in Rare Diseases"
_ICML.cc/2026/Conference — ICML 2026 regular_

### Official Review · Reviewer_J27o · 2026-03-07

**Soundness:** 3
**Presentation:** 2
**Significance:** 3
**Originality:** 2
**Overall Recommendation:** 4
**Confidence:** 3

**Summary:**

Medical diagnosis often requires large-scale data to train reliable models, while in rare disease diagnosis such data is frequently unavailable and the model needs to generalize to unseen tasks. To address this challenge, the paper first uses the DAPE module to refine instructions and obtain an optimal prompt for abnormalities grounding. It then leverages the prior that true abnormalities remain consistent under different perturbations, and proposes V-PUP and RHC to further improve grounding accuracy. Without any training, the proposed DDL framework outperforms both prompt-based methods and fine-tuned models, which suggests strong potential for rare disease grounding.

**Compliance With Llm Reviewing Policy:**

Affirmed.

**Final Justification:**

The author's rebuttal has addressed my concern on efficiency and generalization, which further proves the effectiveness of the paper.

**Key Questions For Authors:**

See the major weaknesses. The core idea is interesting, but the generalization ability and inference time require more detailed clarification and analysis.

**Limitations:**

1. The proposed method substantially improves accuracy at test time, but it requires multiple sampling views, which may increase inference latency largely.
2. The experiments focus mainly on brain MRI, so the generalization to other organs and imaging modalities needs further validation.

**Strengths And Weaknesses:**

Strength:
1. The proposed method achieves even better performance than peft of sft models without any training, which is quite important for rare diseases grounding where data is limited and model's generalization ability to unseen tasks is critical.
2. The paper provides detailed analyses, including thorough ablation studies over different modules and strategies, which supports the effectiveness of the approach.

Major Weakness:
1. The method requires sampling M views at inference. This increases inference time largely and raises concerns about the accuracy-efficiency trade-off. The paper also lacks an analysis of the hyperparameter M. It is strongly recommended to add an experiment that reports both inference time and accuracy across different  M values.
2. The method relies on the assumption that anomalies remain consistent under different perturbations, while hallucinated anomalies are inconsistent. It is unclear whether this assumption is too strong. A visualization study across diverse scenarios is needed to verify whether the assumption holds in practice.
3. In DAPE, the authors generate improved prompts, and this stage is decoupled from the subsequent stages. The paper should therefore test the generated prompts on multiple LLM backbones to evaluate whether the prompts generalize, which will make the contribution more significant.

Minor Issues
1. The size in Table 1 is missing units.
2. The comparative analysis experiment appears only in Table 5. This direct comparison is important and should be moved earlier in the paper.

---

> ### Author Rebuttal · Authors · 2026-03-30
>
> We thank Reviewer J27o for recognizing the importance of abnormality grounding in rare diseases, where labeled data is limited and generalization to unseen tasks is critical. We also appreciate the constructive suggestions and address each point below.
>
> ---
>
> **[Major W1 / Q1] Accuracy-efficiency trade-off**
>
> **We believe DDL has an acceptable accuracy-efficiency trade-off, and most of the added cost can be mitigated by parallelization.** In medical AI, safety is often more critical than latency, making 1.14 s/sample at $M = 7$ acceptable in practice. Performance improves as $M$ increases and begins to saturate around 7, which we therefore use by default. With $M$ GPUs, wall-clock latency can approach single-pass inference (~0.19 s). We therefore believe this cost is practical for clinical workflows.
>
> *Table R1. mAP@50 vs. inference time across M values (Qwen2.5-VL-3B, 100 fixed random samples per dataset).*
>
> | M | NOVA | BTD | Time/sample (serial) | Time/sample (parallel) |
> |---|---|---|---|---|
> | 1 | 0.094 | 0.370 | 0.19s | 0.19s |
> | 3 | 0.122 | 0.378 | 0.49s | ~0.19s |
> | 5 | 0.131 | 0.414 | 0.80s | ~0.19s |
> | 7 | **0.143** | **0.416** | 1.14s | ~0.19s |
>
> ---
>
> **[Major W2] Consistency assumption**
>
> **The consistency assumption is largely supported by the results.** Figure 2 at https://anonymous.4open.science/r/DDL-rebuttal-F805 shows that true abnormalities tend to remain spatially consistent across perturbations, whereas hallucinated or incorrect anchor predictions usually become unstable after perturbation and receive low CRS scores. Table R2 further shows that the CRS score $\sigma$ can serve as a filtering signal to suppress hallucinated detections, where hallucinations are defined as IoU < 0.1. These results support the practical assumption underlying DDL: correct predictions tend to exhibit higher cross-view consensus than hallucinated ones, which is sufficient for ranking and filtering.
>
> *Table R2. CRS thresholding quality and hallucination rate on NOVA.*
>
> | Model | Threshold $\tau$ | Remaining | mAP@50 | Avg IoU | Hallucination Rate ↓ |
> |:---|:---|:---|:---|:---|:---|
> | 3B | $\sigma \ge 0.0$ (no filtering) | 100.0% | 0.150 | 0.231 | 45.1% |
> |  | $\sigma \ge 0.5$ | 88.9% | 0.177 | 0.241 | 43.4% |
> |  | $\sigma \ge 0.7$ | 67.5% | **0.185** | **0.249** | **42.9%** |
> | 7B | $\sigma \ge 0.0$ (no filtering) | 100.0% | 0.206 | 0.254 | 42.0% |
> |  | $\sigma \ge 0.5$ | 69.1% | 0.272 | 0.308 | 35.1% |
> |  | $\sigma \ge 0.7$ | 44.2% | **0.310** | **0.339** | **30.3%** |
>
> ---
>
> **[Major W3] DAPE generalization**
>
> **DAPE is not tied to a particular optimizer LLM or a single target backbone.** As shown in Table R3, all tested optimizers outperform the baseline prompt to varying degrees, and even smaller open-source models provide meaningful gains. Table R4 directly tests prompt transfer across target backbones by evaluating prompts optimized on the 3B backbone on a 7B backbone. The transferred prompts still outperform the vanilla prompt, indicating that the optimized prompts are not merely overfitted to a single backbone and retain transferability across models. Together, these results support the generalization of DAPE across both optimizer LLMs and target backbones.
>
> *Table R3. DAPE across optimizer LLMs on NOVA (mAP@50).*
>
> | Optimizer LLM | Type | 3B Backbone | 7B Backbone |
> |:---|:---|:---|:---|
> | Baseline (No DAPE) | — | 0.088 | 0.135 |
> | **Qwen 3.5-9B** | Open | 0.111 (+26%) | 0.146 (+8%) |
> | **Llama 3.1-8B** | Open | 0.122 (+39%) | 0.147 (+9%) |
> | **GPT-4o** | Closed | 0.129 (+47%) | 0.155 (+15%) |
> | **Gemini 3.0 Flash** | Closed | 0.132 (+50%) | 0.160 (+19%) |
>
> *Table R4. Cross-backbone prompt transfer on NOVA (prompts optimized on 3B and evaluated on 7B).*
>
> | Prompt Source Backbone | Optimizer LLM | Target Backbone | mAP@50 |
> |---|---|---|---|
> | Vanilla | — | 7B | 0.135 |
> | 3B-optimized | GPT-4o | 7B | 0.152 |
> | 3B-optimized | Llama 3.1-8B | 7B | 0.149 |
> | 3B-optimized | Qwen 3.5-9B | 7B | 0.142 |
>
> ---
>
> **Minor issues**
>
> Thank you for pointing out the missing units in Table 1. We will add them in the revision. We also agree that the direct comparative analysis currently shown only in Table 5 is important, and will consider moving it earlier to improve presentation flow.
>
> ---
>
> More broadly, DDL is training-free and architecture-agnostic, without requiring modality-specific adaptation. We focus on brain MRI, and NOVA is, to our knowledge, among the most challenging rare-disease grounding benchmarks, with 281 pathology types, making it a strong dataset for the paper’s central claim. We agree that evaluation on additional organs and imaging modalities would further strengthen the work, and this is a natural next step rather than a limitation of the current contribution.

---

> > ### Author Rebuttal · Reviewer_J27o · 2026-04-03
> >
> > The rebuttal have addressed my concern. I would raise my score to weak accept.

---

> > > ### Author Response · Authors · 2026-04-04
> > >
> > > Thank you very much for your thorough review, especially for emphasizing inference efficiency and prompt generalizability. We also appreciate your helpful suggestions on clarity and presentation, and will incorporate them in the final revision.
> > >
> > > We appreciate the opportunity to clarify these issues through additional experiments during the rebuttal. We are especially grateful that these clarifications addressed your concerns, strengthened the paper, and led you to raise your score and support a weak accept decision.

---

### Official Review · Reviewer_WHTK · 2026-03-13

**Soundness:** 3
**Presentation:** 3
**Significance:** 4
**Originality:** 3
**Overall Recommendation:** 4
**Confidence:** 3

**Summary:**

This paper addresses unstable abnormality grounding in rare-disease scenarios, where data scarcity makes supervised fine-tuning difficult. The authors observe that LVLM grounding is highly sensitive to prompt wording and small visual perturbations, leading to unstable localizations. To address this, the paper proposes Dynamic Decision Learning, which combines instruction-space optimization and visual-consensus verification. The paper also introduces a Consensus Reliability Score to quantify the reliability of predicted grounding boxes. Empirical evaluations are conducted using two clinical brain MRI benchmarks.

**Compliance With Llm Reviewing Policy:**

Affirmed.

**Final Justification:**

The authors have addressed my concerns in the rebuttal, so I will keep my positive recommendation unchanged.

**Key Questions For Authors:**

- Since mAP mainly measures localization accuracy, is it sufficient to guide prompt optimization in DAPE, without explicitly considering robustness or hallucination suppression?
- Are $\omega_1$ and $\omega_2$ fixed across all datasets and model scales, and how sensitive is the method to this choice? If a threshold is applied to the Consensus Reliability Score, can it effectively remove hallucinated detections?
- Since Referenced Hungarian Consolidation uses the original prediction as the reference anchor, can the framework recover when the anchor itself is already incorrect or hallucinated?
- Since DAPE uses Gemini to generate candidate prompts, how sensitive is the method to the choice of the optimizer LLM? In particular, would using a stronger or weaker LLM change the results, and does relying on a strong external LLM make the comparison with baselines less fair?

**Limitations:**

yes

**Strengths And Weaknesses:**

**Strengths**
- The problem of rare disease abnormality grounding to be addressed in this paper is important and of great clinical relevance.
- Overall, the proposed model is novel and technically sound.
- The proposed DAPE is interesting. Instead of using a static instruction, it dynamically optimizes prompts based on high-performing and low-performing instructions.
- The paper also proposes the Consensus Reliability Score to quantify grounding stability across transformed views.
- The experiments are relatively comprehensive, with clear gains over baselines and detailed components of CRS analysis.

**Weaknesses**
- The Consensus Reliability Score seems to act more as a mechanism for evaluating the reliability of predicted bounding boxes, rather than as an explicit module for refining box coordinates.
- Referenced Hungarian Consolidation depends heavily on the reference anchor from the original image, and it is unclear whether the method can recover when the reference anchor itself is incorrect or corresponds to a hallucinated detection.
- The Dynamic Adaptive Prompt Evolution module relies on an external optimizer large language model to generate candidate prompts, which may raise fairness and reproducibility concerns.

---

> ### Author Rebuttal · Authors · 2026-03-30
>
> We thank Reviewer WHTK for the positive assessment of our paper, especially for recognizing its clinical relevance, novelty, and technical soundness, and for the thoughtful questions. We address each point below with additional experiments and clarifications.
>
> ---
>
> **[Q1] Is mAP sufficient for DAPE?**
>
> **Yes, mAP is sufficient for DAPE.** We use mAP to optimize localization quality during prompt selection, while robustness and hallucination suppression are handled by the subsequent visual-consensus stage (V-PUP/RHC). This separation keeps prompt optimization efficient by avoiding multi-view inference at that stage. The ablation supports this design: on NOVA 3B, DAPE alone improves mAP@50 from 0.088 to 0.132, and full DDL further improves it to 0.150 by adding visual verification and consolidation.
>
> ---
>
> **[Q2] Are $w_1$ and $w_2$ fixed across datasets and model scales, and can CRS thresholding reduce hallucinations?**
>
> **Yes. We fix $w_1=0.6$ and $w_2=0.4$ globally across all datasets and backbone sizes, and CRS thresholding can effectively reduce hallucinations.** Here $w_2 = 1 - w_1$. In Table R1, we evaluate whether CRS thresholding can suppress hallucinations (IoU < 0.1). As the table shows, increasing the threshold consistently improves mAP and Avg IoU while reducing the hallucination rate. This suggests that CRS tends to assign lower scores to lower-quality, often hallucinated detections.  In Table R2, we assess the sensitivity to the weighting parameter $w_1$. Different $w_1$ affects the scale of $\sigma$, so threshold-based filtering leads to different performance. Our default choice provides the most balanced results.
>
> *Table R1. CRS thresholding quality and hallucination rate on NOVA (3B and 7B models).*
>
> | Model | Threshold $\tau$ | Remaining | mAP@50 | Avg IoU | Hallucination Rate ↓ |
> |:---|:---|:---|:---|:---|:---|
> | 3B | $\sigma \ge 0.0$ (no filtering) | 100.0% | 0.150 | 0.231 | 45.1% |
> | | $\sigma \ge 0.5$ | 88.9% | 0.177 | 0.241 | 43.4% |
> | | $\sigma \ge 0.7$ | 67.5% | **0.185** | **0.249** | **42.9%** |
> | 7B | $\sigma \ge 0.0$ (no filtering) | 100.0% | 0.206 | 0.254 | 42.0% |
> | | $\sigma \ge 0.5$ | 69.1% | 0.272 | 0.308 | 35.1% |
> | | $\sigma \ge 0.7$ | 44.2% | **0.310** | **0.339** | **30.3%** |
>
> *Table R2. $w_1$ sensitivity on NOVA (mAP@50).*
>
> | Model | Threshold $\tau$ | $w_1=0.2$ | $w_1=0.4$ | $w_1=0.6$ (Default) | $w_1=0.8$ |
> |:---|:---|:---|:---|:---|:---|
> | **3B** | $\sigma \ge 0.0$ | 0.150 | 0.150 | 0.150 | 0.150 |
> | | $\sigma \ge 0.5$ | 0.155 | 0.168 | **0.177** | 0.152 |
> | **7B** | $\sigma \ge 0.0$ | 0.206 | 0.206 | 0.206 | 0.206 |
> | | $\sigma \ge 0.5$ | 0.210 | 0.218 | **0.272** | 0.229 |
>
> ---
>
> **[Q3 / W2] Can DDL recover from an incorrect or hallucinated anchor?**
>
> **DDL can correct anchor errors and identify unreliable anchors.** As shown in the visualization results at https://anonymous.4open.science/r/DDL-rebuttal-F805, RHC can correct anchors with moderate localization errors. When the anchor error is large, DDL does not explicitly regenerate the coordinates. Instead, inconsistent predictions across views usually lead to a low CRS, indicating that the prediction is unreliable.
>
> ---
>
> **[Q4 / W3] Is DAPE sensitive to the optimizer LLM choice?**
>
> **No, DAPE is not sensitive to the optimizer LLM choice.** In Table R3, even the smallest open-source optimizer improves over the baseline. Stronger optimizers do produce slightly better prompts, but the gap between open-source and proprietary models is modest (0.111 vs. 0.132 on 3B). These results suggest that DAPE is not tied to a specific proprietary optimizer LLM, since open-source alternatives also yield consistent gains over the baseline. We therefore view Gemini as a stronger instantiation rather than a prerequisite.
>
> *Table R3. DAPE across optimizer LLMs on NOVA (mAP@50).*
>
> | Optimizer LLM | Type | 3B Backbone | 7B Backbone |
> |:---|:---|:---|:---|
> | Baseline (No DAPE) | — | 0.088 | 0.135 |
> | **Qwen 3.5-9B** | Open | 0.111 (+26%) | 0.146 (+8%) |
> | **Llama 3.1-8B** | Open | 0.122 (+39%) | 0.147 (+9%) |
> | **GPT-4o** | Closed | 0.129 (+47%) | 0.155 (+15%) |
> | **Gemini 3.0 Flash** | Closed | 0.132 (+50%) | 0.160 (+19%) |
>
> ---
>
> **[W1] CRS as evaluation rather than refinement**
>
> **We agree that CRS is used mainly as an analysis signal rather than a direct refinement step, but this score is important because it quantifies prediction reliability.** Our goal is to show that the robustness gain mainly comes from the DDL framework itself, while CRS serves to quantify prediction reliability. As Table R1 suggests, CRS could also be used as a filtering score to further improve results, but we currently present this as an additional benefit rather than a required component of the pipeline.

---

> > ### Author Rebuttal · Reviewer_WHTK · 2026-04-03
> >
> > I appreciate the responses from the authors and my concerns are resolved.

---

> > > ### Author Response · Authors · 2026-04-04
> > >
> > > Thank you very much for your positive assessment and for your helpful questions regarding the design, evaluation, and interpretation of the method. We will incorporate your suggestions to further improve the presentation and discussion in the final version.
> > >
> > > We appreciate the opportunity to clarify these points during the rebuttal and are glad that the additional analyses were helpful in addressing your concerns. We are also grateful that the rebuttal has further strengthened the positive assessment of the paper and reinforced your support for its acceptance. We hope these clarifications are helpful for your final assessment.

---

### Official Review · Reviewer_zKuP · 2026-03-16

**Soundness:** 3
**Presentation:** 2
**Significance:** 3
**Originality:** 2
**Overall Recommendation:** 4
**Confidence:** 3

**Summary:**

The paper proposes Dynamic Decision Learning (DDL), a test-time framework designed to improve abnormality grounding for rare diseases using frozen large vision-language models (LVLMs). Instead of fine-tuning the model, the proposed approach performs iterative refinement at inference time by combining instruction optimization and multi-view visual perturbation verification. Through this process, the model generates multiple predictions, consolidates them to improve localization stability, and derives a consensus-based reliability score that estimates prediction confidence.

**Compliance With Llm Reviewing Policy:**

Affirmed.

**Final Justification:**

Thank you for the detailed and well-structured rebuttal. The additional analyses significantly improve the clarity and strength of the paper. While the technical novelty mainly lies in the integration and test-time formulation rather than a fundamentally new module, I acknowledge that the empirical results and practical relevance (especially for rare-disease scenarios) make the contribution valuable. Overall, my main concerns have been addressed mostly.

**Key Questions For Authors:**

How significant are the improvements across datasets and models?
Some of the reported improvements appear relatively modest. Could the authors provide statistical significance analysis or confidence intervals to better demonstrate the robustness of the gains?

What is the computational overhead of DDL compared to standard inference?
Since the method involves multiple perturbations and iterative refinement, it would be helpful to quantify inference time and computational cost.

Which component contributes the most to the performance improvement?
A more detailed ablation study separating prompt optimization, perturbation consensus, and reliability scoring would help clarify the contribution of each module.

How sensitive is the method to the choice of perturbations and prompts?
Since the framework relies on these mechanisms, it would be useful to understand whether the performance is stable across different perturbation strategies.

**Limitations:**

Partially. The paper briefly discusses limitations related to rare-disease datasets and distribution shifts, but the discussion could be expanded to include computational cost and potential risks of relying on model consensus in clinical decision settings.

**Strengths And Weaknesses:**

# Strengths

1. Practical motivation. The paper addresses an important challenge in medical AI: grounding abnormalities in rare diseases where annotated data is extremely limited. The focus on test-time adaptation without additional training is practically appealing, particularly in medical settings where collecting labeled data is difficult.

2. Model-agnostic design. The proposed framework operates on top of frozen LVLMs and therefore does not require modifying model architectures or retraining large models. This design potentially enables straightforward integration with a wide range of existing vision-language models.

3. Reliability estimation. The use of consensus across perturbations to derive a reliability score is interesting and potentially useful for clinical scenarios where uncertainty estimation is important.

# Weaknesses

1. Limited technical novelty.
The framework largely combines several existing ideas (e.g., prompt optimization, perturbation-based verification, and consensus aggregation). While the integration is reasonable, the paper does not introduce fundamentally new modeling techniques, which somewhat limits the technical depth of the contribution.

2. Improvements may be marginal or dataset-dependent.
Although the paper reports improvements over baselines, it is not entirely clear whether the gains are consistently significant across all settings. In several cases the improvements appear relatively modest, and more detailed statistical analysis or stronger baselines would help clarify whether the improvements are substantial rather than marginal.

3. Limited analysis of computational cost.
The proposed framework involves multiple rounds of inference with perturbations and prompt optimization. However, the paper does not sufficiently discuss the computational overhead or latency implications compared to standard inference. This is particularly important in clinical applications.

4. Presentation issues in figures.
Figures 1 and 2 use a highly stylized and somewhat cartoon-like illustration style. While visually appealing, this style is somewhat unusual for academic publications and may reduce clarity. More conventional schematic diagrams would likely better convey the technical components of the method.

5. Limited component ablation analysis.
While the paper includes some analysis of the DDL components (e.g., the effect of DAPE optimization), the evaluation would benefit from more systematic ablations isolating the contributions of each module (e.g., instruction optimization, visual perturbation verification, and consensus-based reliability scoring). Such analysis would help clarify which component primarily drives the performance gains.

---

> ### Author Rebuttal · Authors · 2026-03-30
>
> We thank Reviewer zKuP for the constructive feedback. We address the concerns below.
>
> **[W1] Technical novelty**
>
> We believe the novelty lies in reframing abnormality grounding with frozen LVLMs as a training-free adaptive inference problem, rather than in any single module. DDL iteratively refines decisions at test time and provides a reliability estimate, beyond prior single-pass grounding or standard fine-tuning. Its gains across datasets and model scales further support its effectiveness, with up to 105% relative improvement in mAP@75 on rare-disease cases over adaptation baselines and supervised fine-tuning. This view is also supported by other reviewers:
> - Reviewer WHTK described DDL as "novel and technically sound" and found DAPE "interesting."
> - Reviewer J27o noted that DDL outperforms other methods without training, highlighting its value in data-limited rare-disease settings.
>
> ---
>
> **[W2/Q1] How significant are the improvements?**
>
> **The gains are large and consistent: averaged over backbones, DDL improves mAP@50 by 43.0% on NOVA and 10.5% on BTD, with clear absolute gains in the main paper.** To verify reliability, we performed Wilcoxon signed-rank tests on mAP@50 between DDL and Vanilla. Across all settings, the gains are statistically significant (all p < 0.05; most p < 0.001), showing that they are consistent rather than noise.
>
> *Table R1. Wilcoxon signed-rank test: DDL vs. Vanilla (mAP@50). Significance levels: * p < 0.05, ** p < 0.01, *** p < 0.001.*
>
> | Dataset | Backbone | p-value | Significance |
> |---------|----------|---------|--------------|
> | NOVA | 3B | 1.14e-10 | *** |
> |  | 7B | 4.55e-13 | *** |
> |  | 32B | 1.24e-03 | ** |
> |  | 72B | 2.08e-05 | *** |
> | BTD | 3B | 4.29e-11 | *** |
> |  | 7B | 3.71e-15 | *** |
> |  | 32B | 3.61e-05 | *** |
> |  | 72B | 9.52e-12 | *** |
>
> ---
>
> **[W3/Q2] Computational overhead**
>
> **We believe DDL has acceptable computational overhead.** In medical AI, safety is often more critical than latency, making 1.14 s/sample at $M = 7$ acceptable in practice. Because each view is independent, DDL is naturally parallelizable. With $M$ GPUs, wall-clock latency can approach single-pass inference (~0.19 s). We therefore believe this cost is practical for clinical workflows.
>
> *Table R2. mAP@50 vs. inference time across M values (Qwen2.5-VL-3B, 100 fixed random samples per dataset).*
>
> | M | NOVA | BTD | Time/sample (serial) | Time/sample (parallel) |
> |---|---|---|---|---|
> | 1 | 0.094 | 0.370 | 0.19s | 0.19s |
> | 3 | 0.122 | 0.378 | 0.49s | ~0.19s |
> | 5 | 0.131 | 0.414 | 0.80s | ~0.19s |
> | 7 | **0.143** | **0.416** | 1.14s | ~0.19s |
>
> ---
>
> **[W5/Q3] Component ablation**
>
> **The full DDL pipeline yields gains beyond prompt optimization alone.** We provide ablations on both datasets. DAPE alone improves 3B NOVA from 0.088 to 0.132 (+50%). Full DDL further improves it to 0.150 (+70%). This shows that V-PUP and RHC provide gains beyond prompt optimization. Similar trends hold across backbones. Aggregation-strategy ablations are shown in Table 2 of the main paper.
>
> *Table R3. Full ablation: mAP@50 across all backbones.*
> | | NOVA | | | BTD | | |
> |---|---|---|---|---|---|---|
> | Backbone | Vanilla | +DAPE | DDL | Vanilla | +DAPE | DDL |
> | 3B | 0.088 | 0.132 | **0.150** | 0.370 | 0.378 | **0.379** |
> | 7B | 0.135 | 0.160 | **0.206** | 0.289 | 0.298 | **0.302** |
> | 32B | 0.208 | 0.224 | **0.266** | 0.367 | 0.422 | **0.433** |
> | 72B | 0.245 | 0.270 | **0.301** | 0.488 | 0.496 | **0.572** |
>
> ---
> **[Q4] Sensitivity to perturbations and prompts**
>
> **DDL is not overly sensitive to the perturbation count or optimizer choice.** Table R2 shows that DDL remains effective across augmentation views $M$. Performance improves as $M$ increases and remains stable even with fewer views, suggesting that DDL does not rely on a narrow choice of $M$.
>
> Table R4 shows that multiple optimizer LLMs, both open-source and closed-source, improve over the vanilla prompt. Table R5 further shows that prompts optimized on 3B remain effective on a 7B backbone, suggesting that the gains are not tied to a single backbone or prompt. These results further support this view.
>
> *Table R4. DAPE across LLM optimizers on NOVA (mAP@50).*
> | Optimizer LLM | Type | 3B | 7B |
> |:---|:---|:---|:---|
> | Vanilla | — | 0.088 | 0.135 |
> | Qwen 3.5-9B | Open | 0.111 | 0.146 |
> | Llama 3.1-8B | Open | 0.122 | 0.147 |
> | GPT-4o | Closed | 0.129 | 0.155 |
> | Gemini 3.0 Flash | Closed | 0.132 | 0.160 |
>
> *Table R5. Cross-backbone prompt transfer on NOVA (prompts optimized on 3B and evaluated on 7B).*
> | Prompt Source Backbone | Optimizer LLM | Target Backbone | mAP@50 |
> |---|---|---|---|
> | Vanilla | — | 7B | 0.135 |
> | 3B-optimized | GPT-4o | 7B | 0.152 |
> | 3B-optimized | Llama 3.1-8B | 7B | 0.149 |
> | 3B-optimized | Qwen 3.5-9B | 7B | 0.142 |
>
> ---
>
> **[W4] Figure style**
>
> We appreciate this suggestion and have removed the cartoon-style presentation from Figure 1. The updated figure is available here: https://anonymous.4open.science/r/DDL-rebuttal-F805 .

---

> > ### Author Rebuttal · Reviewer_zKuP · 2026-04-03
> >
> > Thank you for the detailed and well-structured rebuttal. The additional analyses significantly improve the clarity and strength of the paper. While the technical novelty mainly lies in the integration and test-time formulation rather than a fundamentally new module, I acknowledge that the empirical results and practical relevance (especially for rare-disease scenarios) make the contribution valuable. Overall, my main concerns have been addressed mostly.

---

> > > ### Author Response · Authors · 2026-04-04
> > >
> > > Thank you very much for your thorough review and for raising insightful questions regarding statistical significance, efficiency trade-offs, and component-level contributions. We will also incorporate your presentation suggestions in the final version.
> > >
> > > We are glad that the additional experiments provided during the rebuttal sufficiently addressed your concerns. We are especially grateful that these clarifications helped strengthen the paper and led you to support a weak accept decision with an increased score.

---

### Decision · Program_Chairs · 2026-04-30

**Decision:**

Accept (regular)

**Comment:**

All reviewers gave positive scores and the authors' rebuttals solve the raised problems well.